# LATENT IMAGE ANIMATOR:
# LEARNING TO ANIMATE IMAGES VIA LATENT SPACE NAVIGATION

**Yaohui Wang, Di Yang, Francois Bremond & Antitza Dantcheva**
Inria, Université Côte d'Azur
{yaohui.wang,di.yang,francois.bremond,antitza.dantcheva}@inria.fr

## ABSTRACT

Due to the remarkable progress of deep generative models, animating images has become increasingly efficient, whereas associated results have become increasingly realistic. Current animation-approaches commonly exploit structure representation extracted from driving videos. Such structure representation is instrumental in transferring motion from driving videos to still images. However, such approaches fail in case the source image and driving video encompass large appearance variation. Moreover, the extraction of structure information requires additional modules that endow the animation-model with increased complexity. Deviating from such models, we here introduce the Latent Image Animator (LIA), a self-supervised autoencoder that evades need for structure representation. LIA is streamlined to animate images by linear navigation in the latent space. Specifically, motion in generated video is constructed by linear displacement of codes in the latent space. Towards this, we learn a set of orthogonal motion directions simultaneously, and use their linear combination, in order to represent any displacement in the latent space. Extensive quantitative and qualitative analysis suggests that our model systematically and significantly outperforms state-of-art methods on VoxCeleb, Taichi and TED-talk datasets *w.r.t.* generated quality. Source code and pre-trained models are publicly available[1].

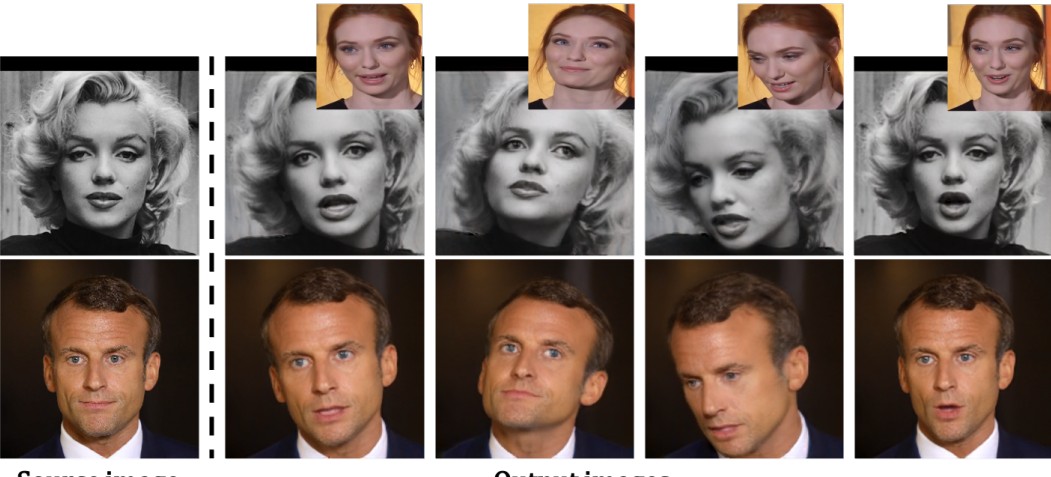

Source image                    Output images

Figure 1: **LIA animation examples.** The two images of Marilyn Monroe and Emmanuel Macron are animated by LIA, which transfers motion of a driving video (smaller images on the top) from VoxCeleb dataset (Chung et al., 2018) onto the still images. LIA is able to successfully animate these two images without relying on any *explicit structure representations*, such as landmarks and region representations.

---

[1]https://wyhsirius.github.io/LIA-project/

# 1 INTRODUCTION

In the series of science fiction books *Harry Potter* (Rowling et al., 2016; Rowling, 2019), wizards and witches were able to magically *enchant portraits, bringing them to life*. Remarkable progress of deep generative models has recently turned this vision into reality. This work examines the scenario where a framework animates a *source image* by motion representations learned from a *driving video*. Existing approaches for image animation are classically related to computer graphics (Cao et al., 2014; Thies et al., 2016; 2019; Zhao et al., 2018) or exploit motion labels (Wang et al., 2020b) and structure representations such as semantic maps (Pan et al., 2019; Wang et al., 2018; 2019), human keypoints (Jang et al., 2018; Yang et al., 2018; Walker et al., 2017; Chan et al., 2019; Zakharov et al., 2019; Wang et al., 2019; Siarohin et al., 2019), 3D meshes (Liu et al., 2019; Chen et al., 2021), and optical flows (Li et al., 2018; Ohnishi et al., 2018). We note that the ground truth of such structure representations has been computed a-priori for the purpose of supervised training, which poses constraints on applications, where such representations of unseen testing images might be fragmentary or difficult to access.

Self-supervised motion transfer approaches (Wiles et al., 2018; Siarohin et al., 2019; 2021) accept raw videos as input and learn to reconstruct driving images by warping source image with predicted *dense optical flow fields*. While the need for domain knowledge or labeled ground truth data has been obviated, which improves performance on in-the-wild testing images, such methods entail necessity of explicit structure representations as motion guidance. Prior information such as keypoints (Siarohin et al., 2019; Wang et al., 2021a) or regions (Siarohin et al., 2021) are learned in an end-to-end training manner by additional networks as intermediate features, in order to predict target flow fields. Although online prediction of such representations is less tedious than the acquisition of ground truth labels, it still strains the complexity of networks.

Deviating from such approaches, we here aim to fully *eliminate* the need of *explicit structure representations* by directly manipulating the latent space of a deep generative model. To the best of our knowledge, this constitutes a new direction in the context of *image animation*. Our work is motivated by *interpretation of GANs* (Shen et al., 2020; Goetschalckx et al., 2019; Jahanian et al., 2020; Voynov & Babenko, 2020), showcasing that latent spaces of StyleGAN (Karras et al., 2019; 2020b) and BigGAN (Brock et al., 2019) contain rich semantically meaningful directions. Given that walking along such directions, basic visual transformations such as *zooming* and *rotation* can be induced in generated results. As in image animation, we have that motion between source and driving images can be considered as higher-level transformation, a natural question here arises: *can we discover a set of directions in the latent space that induces high-level motion transformations collaboratively?*

Towards answering this question, we introduce LIA, a novel Latent Image Animator constituting of an autoencoder for animating still images via latent space navigation. LIA seeks to animate a source image via linearly navigating associated source latent code along a learned path to reach a target latent code, which represents the high-level transformation for animating the source image. We introduce a Linear Motion Decomposition (LMD) approach aiming to represent a latent path via a linear combination of a set of learned motion directions and associated magnitudes. Specifically, we constrain the set as an orthogonal basis, where each vector indicates a basic visual transformation. By describing the whole motion space using such learned basis, LIA eliminates the requirement of explicit structure representations.

In addition, we design LIA to disentangle motion and appearance within a single encoder-generator architecture. Deviating from existing methods using separate networks to learn disentangled features, LIA integrates both, latent *motion* code, as well as *appearance* features in a *single encoder*, which highly reduces the model complexity and simplifies training.

We provide evaluation on multiple datasets including VoxCeleb (Chung et al., 2018), TaichiHD (Siarohin et al., 2019) and TED-talk (Siarohin et al., 2021). In addition, we show that LIA outperforms the state-of-the-art in preserving the facial structure in generated videos in the setting of one-shot image animation on unseen datasets such as FFHQ (Karras et al., 2019) and GermanPublicTV (Thies et al., 2020).

# 2 RELATED WORK

**Video generation** GAN-based video generation is aimed at mapping Gaussian noise to video, directly and in the absence of prior information (Vondrick et al., 2016; Saito et al., 2017; Tulyakov et al., 2018; Wang et al., 2020a; Wang, 2021). Approaches based on deep probabilistic models (Den-

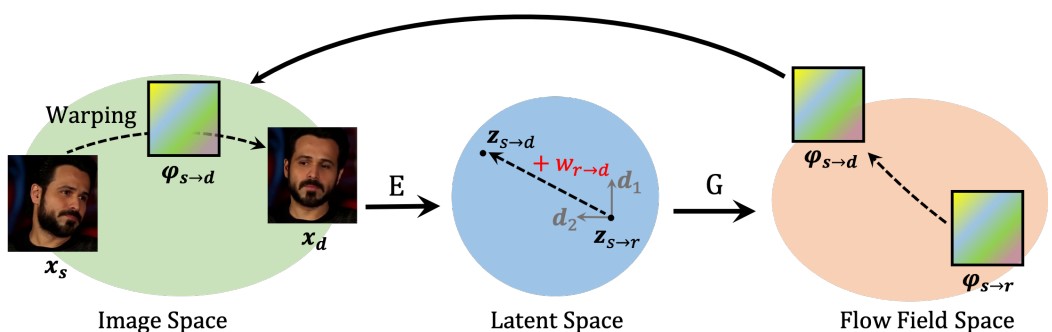

Figure 2: **General pipeline.** Our objective is to transfer motion via latent space navigation. The entire training pipeline consists of two steps. Firstly, we encode a source image $x_s$ into a latent code $z_{s \to r}$. By linearly navigating $z_{s \to r}$ along a path $w_{r \to d}$, we reach a target latent code $z_{s \to d}$. The latent paths are represented by a linear combination between a set of learned motion directions (*e.g.*, $d_1$ and $d_2$), which is an orthogonal basis, and associated magnitudes. In the second step, we decode $z_{s \to d}$ to a target dense optical flow field $\phi_{s \to d}$, which is used to warp $x_s$ into the driving image $x_d$. While we train our model using images from the same video sequence, in the testing phase, $x_s$ and $x_d$ generally pertain to different identities.

ton & Birodkar, 2017; Li & Mandt, 2018; Bhagat et al., 2020; Xie et al., 2020) were also proposed to tackle this problem, however only show results on toy datasets with low resolution. Recently, with the progress of GANs in photo-realistic image generation (Brock et al., 2019; Karras et al., 2019; 2020a), a series of works (Clark et al., 2019; Wang et al., 2021c) explored production of high-resolution videos by incorporating the architecture of an image generator into video GANs, trained jointly with RNNs. Tian et al. (2021) directly leveraged the knowledge of a pre-trained StyleGAN to produce videos of resolution up to $1024 \times 1024$. Unlike these approaches, which generate random videos based on noise vectors in an unconditional manner, in this paper, we focus on conditionally creating novel videos by transferring motion from driving videos to input images.

**Latent space editing** In an effort to control generated images, recent works explored the discovery of semantically meaningful directions in the latent space of pre-trained GANs, where linear navigation corresponds to desired image manipulation. Supervised (Shen et al., 2020; Jahanian et al., 2020; Goetschalckx et al., 2019) and unsupervised (Voynov & Babenko, 2020; Peebles et al., 2020; Shen & Zhou, 2021) approaches were proposed to edit semantics such as facial attributes, colors and basic visual transformations (*e.g.,* rotation and zooming) in generated or inverted real images (Zhu et al., 2020; Abdal et al., 2020). In this work, as opposed to finding directions corresponding to individual visual transformations, we seek to learn a set of directions that cooperatively allows for high-level visual transformations that can be beneficial in image animation.

**Image animation** Related approaches (Chan et al., 2019; Wang et al., 2018; Zakharov et al., 2019; Wang et al., 2019; Yang et al., 2020) in image animation required strong prior structure labels as motion guidance. In particular, Chan et al. (2019), Yang et al. (2020) and Wang et al. (2018) proposed to map representations such as human keypoints and facial landmarks to videos in the setting of image-to-image translation proposed by Isola et al. (2017). However, such approaches were only able to learn an individual model for a single identity. Transferring motion on new appearances requires retraining the entire model from scratch by using videos of target identities. Several recent works (Zakharov et al., 2019; Wang et al., 2019) explored meta learning in fine-tuning models on target identities. While only few images of target identities were required during inference time, it was still compulsory to input pre-computed structure representations in those approaches, which usually are hard to access in many real-world scenarios. Towards addressing this issue, very recent works (Siarohin et al., 2019; 2021; Wang et al., 2021b; Wiles et al., 2018) proposed to learn image animation in self-supervised manner, only relying on RGB videos for both, training and testing without any priors. They firstly predicted dense flow fields from input images, which were then utilized to warp source images, in order to obtain final generated results. Inference only required one image of a target identity without any fine-tuning step on pre-trained models. While no priors were required, state-of-the-art methods still followed the idea of using explicit structure representations. FOMM (Siarohin et al., 2019) proposed a first order motion approach to predict keypoints and local transformations online to generate flow fields. Siarohin et al. (2021) developed this idea to model articulated objects by replacing a keypoints predictor by a PCA-based region prediction module. Wang et al. (2021b) extended FOMM by predicting 3D keypoints for view-free generation. We

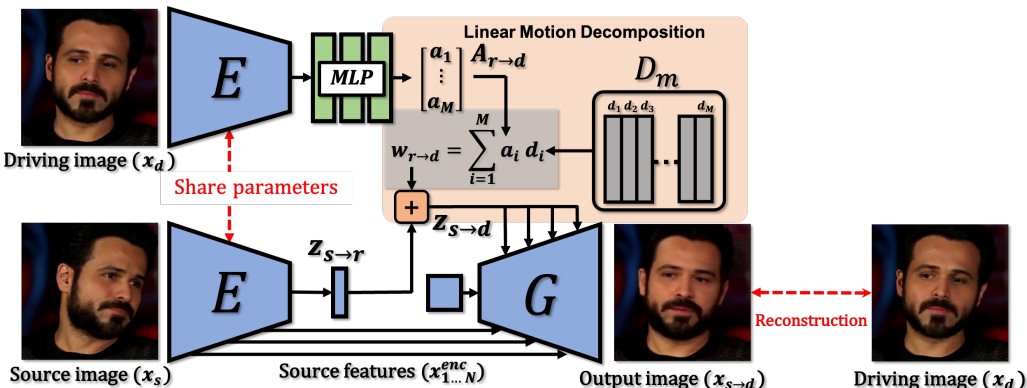

Figure 3: **Overview of LIA.** LIA is an autoencoder consisting of two networks, an encoder $E$ and a generator $G$. In the latent space, we apply Linear Motion Decomposition (LMD) towards learning a motion dictionary $D_m$, which is an orthogonal basis where each vector represents a basic visual transformation. LIA takes two frames sampled from the same video sequence as source image $x_s$ and driving image $x_d$ respectively during training. Firstly, it encodes $x_s$ into a source latent code $z_{s \to r}$ and $x_d$ into a magnitude vector $A_{r \to d}$. Then, it linearly combines $A_{r \to d}$ and a trainable $D_m$ using LMD to obtain a latent path $w_{r \to d}$, which is used to navigate $z_{s \to r}$ to a target code $z_{s \to d}$. Finally, $G$ decodes $z_{s \to d}$ into a target dense flow field and warps $x_s$ to an output image $x_{s \to d}$. The training objective is to reconstruct $x_d$ using $x_{s \to d}$.

note though that in all approaches, given that keypoints or regions are inadequately predicted, the quality of generated images drastically decreases. In contrast to such approaches, our method does not require any explicit structure representations. We dive into the latent space of the generator and self-learn to navigate motion codes in certain directions with the goal to reach target codes, which are then decoded to flow fields for warping.

## 3 METHOD

Self-supervised image animation aims at learning to transfer motion from a subject of a driving video to a subject in a source image based on training with a large video dataset. In this work, we propose to model such motion transformation via latent space navigation. The general pipeline is illustrated in Fig. 2. Specifically, for training, our model takes in a pair of source and driving images, randomly sampled from one video sequence. These two images are encoded into a latent code which is used to represent motion transformation in the image space. The training objective is to reconstruct the driving image by combining source image with learned motion transformation. For testing, frames of a driving video are sequentially processed with the source image to animate the source subject.

We provide an overview of the proposed model in Fig. 3. Our model is an autoencoder, consisting of two main networks, an encoder $E$ and a generator $G$. In general, our model requires two steps to transfer motion. In the first step, $E$ encodes source and driving images $x_s, x_d \sim \mathcal{X} \in \mathbb{R}^{3 \times H \times W}$ into latent codes in the latent space. The source code is then navigated into a target code, which is used to represent target motion transformation, along a learned latent path. Based on proposed Linear Motion Decomposition (LMD), we represent such a path as a linear combination of a set of learned motion directions and associated magnitudes, which are learned from $x_d$. In the second step, once the target latent code is obtained, $G$ decodes it as a dense flow field $\phi_{s \to d} \sim \Phi \in \mathbb{R}^{2 \times H \times W}$ and uses $\phi_{s \to d}$ to warp $x_s$ and then to obtain the output image. In the following, we proceed to discuss the two steps in detail.

### 3.1 LATENT MOTION REPRESENTATION

Given a source image $x_s$ and a driving image $x_d$, our first step constitutes of learning a *latent code* $z_{s \to d} \sim \mathcal{Z} \in \mathbb{R}^N$ to represent the motion transformation from $x_s$ to $x_d$. Due to the uncertainty of two images, directly learning $z_{s \to d}$ puts forward a high requirement on the model to capture a complex distribution of motion. Mathematically, it requires modeling directions and norms of the vector $z_{s \to d}$ simultaneously, which is challenging. Therefore, instead of modeling motion transformation $x_s \to x_d$, we assume there exists a reference image $x_r$ and motion transfer can be modeled

as $x_s \rightarrow x_r \rightarrow x_d$, where $z_{s \rightarrow d}$ is learned in an indirect manner. We model $z_{s \rightarrow d}$ as a target point in the latent space, which can be reached by taking linear walks from a starting point $z_{s \rightarrow r}$ along a linear path $w_{r \rightarrow d}$ (see Fig. 2), given by

$$z_{s \rightarrow d} = z_{s \rightarrow r} + w_{r \rightarrow d}, \tag{1}$$

where $z_{s \rightarrow r}$ and $w_{r \rightarrow d}$ indicate the transformation $x_s \rightarrow x_r$ and $x_r \rightarrow x_d$ respectively. Both $z_{s \rightarrow r}$ and $w_{r \rightarrow d}$ are learned independently and $z_{s \rightarrow r}$ is obtained by passing $x_s$ through $E$.

We learn $w_{r \rightarrow d}$ via Linear Motion Decomposition (LMD). Our idea is to learn a set of motion directions $D_m = \{\mathbf{d_1}, ..., \mathbf{d_M}\}$ to represent any path in the latent space. We constrain $D_m$ as an orthogonal basis, where each vector indicates a motion direction $\mathbf{d_i}$. We then combine each vector in the basis with a vector $A_{r \rightarrow d} = \{a_1, ..., a_M\}$, where $a_i$ represents the magnitude of $\mathbf{d_i}$. Hence, any linear path in the latent space can be represented using a linear combination

$$w_{r \rightarrow d} = \sum_{i=1}^{M} a_i \mathbf{d_i}, \tag{2}$$

where $\mathbf{d_i} \in \mathbb{R}^N$ and $a_i \in \mathbb{R}$ for all $i \in \{1, ..., M\}$. Semantically, each $\mathbf{d_i}$ should represent a basic visual transformation and $a_i$ indicates the required steps to walk in $\mathbf{d_i}$ towards achieving $w_{r \rightarrow d}$. Due to $D_m$ entailing an orthogonal basis, any two directions $\mathbf{d_i}, \mathbf{d_j}$ follow the constrain

$$< \mathbf{d_i}, \mathbf{d_j} > = \begin{cases} 0 & i \neq j \\ 1 & i = j. \end{cases} \tag{3}$$

We implement $D_m$ as a learnable matrix and apply the Gram-Schmidt process during each forward pass, in order to meet the requirement of orthogonality. $A_{r \rightarrow d}$ is obtained by mapping $z_{d \rightarrow r}$, which is the output of $x_d$ after $E$, through a 5-layer MLP. The final formulation of latent motion representation for each $x_s$ and $x_d$ is thus given as

$$z_{s \rightarrow d} = z_{s \rightarrow r} + \sum_{i=1}^{M} a_i \mathbf{d_i}. \tag{4}$$

### 3.2 Latent code driven image animation

Once we obtain $z_{s \rightarrow d}$, in our second step, we use $G$ to decode a flow field $\phi_{s \rightarrow d}$ and warp $x_s$. Our $G$ consists of two components, a flow field generator $G_f$ and a refinement network $G_r$ (we provide details in App. A).

Towards learning multi-scale features, $G$ is designed as a residual network containing $N$ models to produce a pyramid of flow fields $\phi_{s \rightarrow d} = \{\phi_i\}_1^N$ in different layers of $G_f$. Multi-scale source features $x_s^{enc} = \{x_i^{enc}\}_1^N$ are obtained from $E$ and are warped in $G_f$.

However, as pointed out by Siarohin et al. (2019), only relying on $\phi_{s \rightarrow d}$ to warp source features is insufficient to precisely reconstruct driving images due to the existing occlusions in some positions of $x_s$. In order to predict pixels in those positions, the network is required to inpaint the warped feature maps. Therefore, we predict multi-scale masks $\{m_i\}_1^N$ along with $\{\phi_i\}_1^N$ in $G_f$ to mask out the regions required to be inpainted. In each residual module, we have

$$x_i' = \mathcal{T}(\phi_i, x_i^{enc}) \odot m_i, \tag{5}$$

where $\odot$ denotes the Hadamard product and $\mathcal{T}$ denotes warping operation, whereas $x_i'$ signifies the masked features. We generate both dense flow fields, as well as masks by letting each residual module output a 3-channel feature map in which the first two channels represent $\phi_i$ and the last channel $m_i$. Based on an inpainted feature map $f(x_i')$, as well as an upsampled image $g(x_{i-1})$ provided by the previous module in $G_r$, the RGB image from each module is given by

$$o_i = f(x_i') + g(o_{i-1}), \tag{6}$$

where $f$ and $g$ denote the inpainting and upsampling layers, respectively. The output image $o_N$ from the last module constitutes the final generated image $x_{s \rightarrow d} = o_N$.

### 3.3 Learning

We train LIA in a self-supervised manner to reconstruct $x_d$ using three losses, *i.e.*, a reconstruction loss $\mathcal{L}_{recon}$, a perceptual loss $\mathcal{L}_{vgg}$ (Johnson et al., 2016) and an adversarial loss $\mathcal{L}_{adv}$. We use $\mathcal{L}_{recon}$ to minimize the pixel-wise $L_1$ distance between $x_d$ and $x_{s \rightarrow d}$, calculated as

$$\mathcal{L}_{recon}(x_{s \rightarrow d}, x_d) = \mathbb{E}[\|x_d - x_{s \rightarrow d}\|_1]. \tag{7}$$

Towards minimizing the perceptual distance, we apply a VGG19-based $\mathcal{L}_{vgg}$ on multi-scale feature maps between real and generated images, written as

$$\mathcal{L}_{vgg}(x_{s \to d}, x_d) = \mathbb{E}[\sum_{n}^{N} \|F_n(x_d) - F_n(x_{s \to d})\|_1], \tag{8}$$

where $F_n$ denotes the $n^{th}$ layer in a pre-trained VGG19 (Simonyan & Zisserman, 2015). In practice, towards penalizing real and generated images in multi-scale images, we use a pyramid of four resolutions, namely $256 \times 256$, $128 \times 128$, $64 \times 64$ and $32 \times 32$ as inputs of VGG19. The final perceptual loss is the addition of perceptual losses in four resolutions.

Further, towards generating photo-realistic results, we incorporate a non-saturating adversarial loss $\mathcal{L}_{adv}$ on $x_{s \to d}$, which is calculated as

$$\mathcal{L}_{adv}(x_{s \to d}) = \mathbb{E}_{x_{s \to d} \sim p_{rec}}[-log(D(x_{s \to d}))], \tag{9}$$

where $D$ is a discriminator, aimed at distinguishing reconstructed images from the original ones. Our full loss function is the combination of three losses with $\lambda$ as a balanced hyperparameter

$$\mathcal{L}(x_{s \to d}, x_d) = \mathcal{L}_{recon}(x_{s \to d}, x_d) + \lambda \mathcal{L}_{vgg}(x_{s \to d}, x_d) + \mathcal{L}_{adv}(x_{s \to d}). \tag{10}$$

### 3.4 INFERENCE

In inference stage, given a driving video sequence $V_d = \{x_t\}_1^T$, we aim to transfer motion from $V_d$ to $x_s$, in order to generate a novel video $V_{d \to s} = \{x_{t \to s}\}_1^T$. If $V_d$ and $x_s$ stem from the same video sequence, *i.e.*, $x_s = x_1$, our task comprises of reconstructing the entire original video sequence. Therefore, we construct the latent motion representation of each frame using *absolute transfer*, which follows the training process, given as

$$z_{s \to t} = z_{s \to r} + w_{r \to t}, \ t \in \{1, ..., T\}. \tag{11}$$

However, in real world applications, interest is rather placed on the scenario, where motion transfer between $x_s$ and $V_d$, the latter stemming from different identities, *i.e.*, $x_s \neq x_1$. Taking a *talking head* video as an example, in this setting, beyond identity, $x_1$ and $x_s$ might also differ in pose and expression. Therefore, we propose *relative transfer* to eliminate the motion impact of $w_{r \to 1}$ and involve motion of $w_{r \to s}$ in the full generated video sequence. Owing to a linear representation of the latent path, we can easily represent $z_{s \to t}$ for each frame as

$$
\begin{aligned}
z_{s \to t} &= (z_{s \to r} + w_{r \to s}) + (w_{r \to t} - w_{r \to 1}) \\
&= z_{s \to s} + (w_{r \to t} - w_{r \to 1}), \ t \in \{1, ..., T\}.
\end{aligned} \tag{12}
$$

The first term in Eq. (12), $z_{s \to s}$ indicates the reconstruction of $x_s$, while the second term $(w_{r \to t} - w_{r \to 1})$ represents the motion from $x_1$ to $x_t$. This equation indicates that the original pose is preserved in $x_s$, at the same time motion is transferred from $V_d$. We note that in order to completely replicate the position and pose in $V_d$, it requires $x_s$ and $x_1$ to contain similar poses in relative motion transfer.

## 4 EXPERIMENTS

In this section, we firstly describe our experimental setup including implementation details and datasets. Secondly, we qualitatively demonstrate generated results based on testing datasets. Then, we provide quantitative evaluation *w.r.t.* image quality on (a) same-identity reconstruction, (b) cross-video motion transfer, presenting (c) a user study. Next, we conduct an ablation study that demonstrates (d) the effectiveness of our proposed motion dictionary, as well as (e) associated size. Finally, we provide an in-depth analysis of our (f) latent codes and (g) motion dictionary to interpret their semantic meanings.

**Datasets** Our model is trained on the datasets VoxCeleb, TaichiHD and TED-talk. We follow the pre-processing method in (Siarohin et al., 2019) to crop frames into $256 \times 256$ resolution for quantitative evaluation.

**Implementation details** Our model is implemented in PyTorch (Paszke et al., 2019). All models are trained on four 16G NVIDIA V100 GPUs. The total batch size is 32 with 8 images per GPU. We use a learning rate of 0.002 to train our model with the Adam optimizer (Kingma & Ba, 2014). The dimension of all latent codes, as well as directions in $D_m$ is set to be 512. In our loss function, we use $\lambda = 10$ in order to penalize more on the perceptual loss. It takes around 150 hours to fully train our framework.

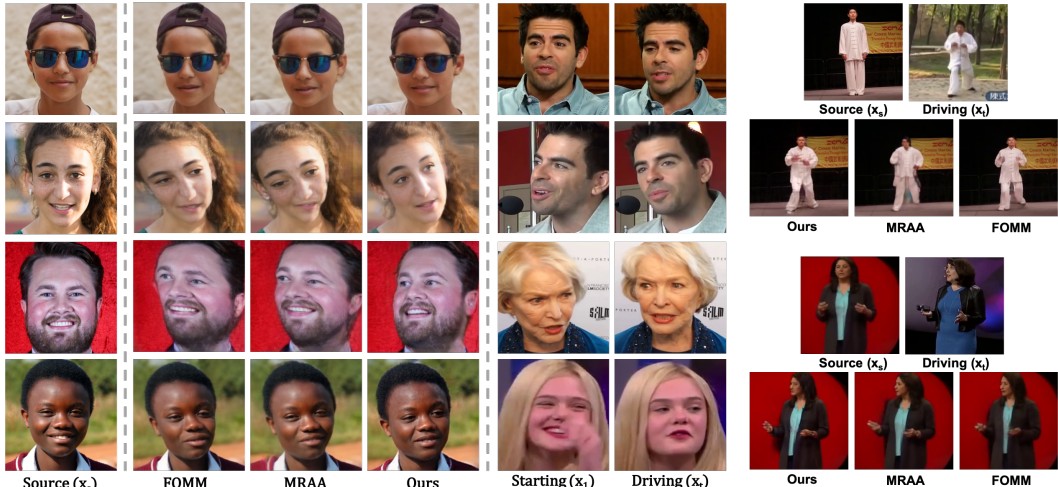

Figure 4: **Qualitative results.** Examples for same-dataset *absolute motion transfer* on TaichiHD (top-right) and TED-talk (bottom-right). On VoxCeleb (left), we demonstrate cross-dataset *relative motion transfer*. We successfully transfer motion between $x_1$ and $x_t$ from videos in VoxCeleb to $x_s$ from FFHQ, the latter not being used for training.

**Evaluation metrics** We evaluate our model *w.r.t.* (i) reconstruction faithfulness using $\mathcal{L}_1$, LPIPS, (ii) generated video quality using video FID, as well as (iii) semantic consistency using average keypoint distance (AKD), missing keypoint rate (MKR) and average euclidean distance (AED). Details are available in App. B.2.

## 4.1 QUALITATIVE RESULTS

Firstly, we evaluate the ability of LIA to generate realistic videos and compare related results with four state-of-the-art methods. For TaichiHD and TED-talk datasets, we conduct an experiment related to cross-video generation. Corresponding results (see Fig. 4) confirm that our method is able to correctly transfer motion on articulated human bodies, in the absence of explicit structure representations. For the VoxCeleb dataset, we conduct a cross-dataset generation-experiment, where we transfer motion from VoxCeleb to images of the FFHQ dataset. We observe that our method outperforms FOMM and MRAA *w.r.t.* image quality, as both approaches visibly deform the shape of the original faces. This is specifically notable in the case that source and driving images entail large pose variations. At the same time, LIA is able to successfully tackle this challenge and no similar deformations are visible.

## 4.2 COMPARISON WITH STATE-OF-THE-ART METHODS

We quantitatively compare our method with the state-of-the-art approaches X2Face, Monkey-Net, FOMM and MRAA on two tasks, namely (a) same-identity reconstruction and (b) cross-video motion transfer. Additionally, we conduct a (c) user study.

**(a) Same-identity reconstruction** We here evaluate the reconstruction ability of our method. Specifically, we reconstruct each testing video by using the first frame as $x_s$ and the remaining frames as $x_d$. Results on three datasets are reported in Table 1. Focusing on foreground-reconstruction, our method outperforms the other approaches *w.r.t.* all metrics. More results are presented in App. B.3, discussing background-reconstruction.

**(b) Cross-video motion transfer** Next, we conduct experiments, where source images and driving videos stem from different video sequences. In this context, we mainly focus on evaluating *talking head* videos and explore two different cases. In the first case, we generate videos using the VoxCeleb testing set to conduct *motion transfer*. In the second case, source images are from an unseen dataset, namely the GermanPublicTV dataset, as we conduct *cross-dataset motion transfer*. In both experiments, we randomly construct source and driving pairs and transfer motion from driving videos to source images to generate a novel manipulated dataset. Since ground truth data for our generated videos is not available, we use video FID (as initialized by Wang et al. (2020a)) to

Table 1: **Same-identity reconstruction.** Comparison with state-of-the-art methods on three datasets for same-identity reconstruction.

| | VoxCeleb | | | | TaichiHD | | | | TED-talks | | | |
|---|---|---|---|---|---|---|---|---|---|---|---|---|
| Method | $\mathcal{L}_1$ | AKD | AED | LPIPS | $\mathcal{L}_1$ | (AKD, MKR) | AED | LPIPS | $\mathcal{L}_1$ | (AKD, MKR) | AED | LPIPS |
| X2Face | 0.078 | 7.687 | 0.405 | - | 0.080 | (17.654, 0.109) | - | - | - | - | - | - |
| Monkey-Net | 0.049 | 1.878 | 0.199 | - | 0.077 | (10.798, 0.059) | - | - | - | - | - | - |
| FOMM | 0.046 | 1.395 | 0.141 | 0.136 | 0.063 | (6.472, 0.032) | 0.495 | 0.191 | 0.030 | (3.759, 0.0090) | 0.428 | 0.13 |
| MRAA *w/o* bg | 0.043 | **1.307** | 0.140 | 0.127 | 0.063 | (5.626, 0.025) | 0.460 | 0.189 | 0.029 | (**3.126, 0.0092**) | **0.396** | 0.12 |
| Ours | **0.041** | 1.353 | **0.138** | **0.123** | **0.057** | (**4.823, 0.020**) | **0.431** | **0.180** | **0.027** | (3.141, 0.0095) | 0.399 | **0.11** |

Table 2: **Cross-video generation.** We report video FID for both inner- and cross-dataset tasks on VoxCeleb and GermanPublicTV.

| | VoxCeleb | GermanPublicTV |
|---|---|---|
| FOMM | 0.323 | 0.456 |
| MRAA | 0.308 | 0.454 |
| Ours | **0.161** | **0.406** |

Table 3: **User study.** We ask 20 human raters to conduct a subjective video quality evaluation.

| | VoxCeleb(%) | TaichiHD(%) | TED-talk(%) |
|---|---|---|---|
| Ours/FOMM | **92.9**/7.1 | **64.5**/35.5 | **71.4**/28.6 |
| Ours/MRAA | **89.7**/10.3 | **60.7**/39.9 | **54.8**/45.2 |

compute the distance between generated and real data distributions. As shown in Tab. 2, our method outperforms all other approaches *w.r.t.* video FID, indicating the best generated video quality.

**(c) User study** We conduct a user study to evaluate video quality. Towards this, we displayed paired videos and asked 20 human raters 'which clip is more realistic?'. Each video-pair contains a generated video from our method, as well as a video generated from FOMM or MRAA. Results suggest that our results are more realistic in comparison to FOMM and MRAA across all three datasets (see Tab. 3). Hence, the obtained human preference is in accordance with our quantitative evaluation.

Table 4: **Ablation study on motion dictionary.** We conduct experiments on three datasets with and without $D_m$ and show reconstruction results.

| | VoxCeleb | | TaichiHD | | TED-talks | |
|---|---|---|---|---|---|---|
| Method | $\mathcal{L}_1$ | LPIPS | $\mathcal{L}_1$ | LPIPS | $\mathcal{L}_1$ | LPIPS |
| w/o $D_m$ | 0.049 | 0.165 | 0.062 | 0.186 | 0.031 | 0.12 |
| Full | **0.041** | **0.123** | **0.057** | **0.180** | **0.028** | **0.11** |

Table 5: **Ablation study on $D_m$ size.** We conduct experiments on three datasets with 5 different $D_m$ size and show reconstruction results.

| | VoxCeleb | | TaichiHD | | TED-talks | |
|---|---|---|---|---|---|---|
| M | $\mathcal{L}_1$ | LPIPS | $\mathcal{L}_1$ | LPIPS | $\mathcal{L}_1$ | LPIPS |
| 5 | 0.051 | 0.15 | 0.070 | 0.22 | 0.037 | 0.15 |
| 10 | 0.043 | 0.13 | 0.065 | 0.20 | 0.036 | 0.13 |
| 20 | **0.041** | **0.12** | **0.057** | **0.18** | **0.028** | **0.11** |
| 40 | 0.042 | **0.12** | 0.060 | 0.19 | 0.030 | 0.12 |
| 100 | **0.041** | **0.12** | 0.058 | **0.18** | **0.028** | **0.11** |

### 4.3 ABLATION STUDY

We here analyze our proposed motion dictionary and focus on answering following two questions.

**(d) Is the motion dictionary $D_m$ beneficial?** We here explore the impact of proposed $D_m$, by training our model without $D_m$. Specifically, we output $w_{r \to d}$ directly from MLP, without using LMD to learn an orthogonal basis. From the evaluation results reported in Tab. 4 and qualitative results in App. B.5, we observe that in the absence of $D_m$, model fails to generate high-quality images, which proves the effectiveness of $D_m$, consistently on all datasets.

**(e) How many directions are required in $D_m$?** Towards finding an effective size of $D_m$, we empirically test three different $M$, viz. 5, 10, 20, 40 and 100. Quantitative results in Tab. 5 show that when using 20 directions, the model achieves the best reconstruction results.

### 4.4 FURTHER ANALYSIS

**(f) Latent code analysis.** While our method successfully transfers motion via latent space navigation, we here aim at answering the question — *what does $x_r$ represent*? Towards answering this question, we proceed to visualize $x_r$. We firstly decode $z_{s \to r}$ into a dense flow field $\phi_{s \to r}$, which is then used to warp $x_s$ (we show details in App. B.4). Fig. 5 shows examples of $x_s$ and $x_r$. Interestingly, we observe that $x_r$ represents the *canonical pose* of $x_s$, regardless of original poses of the subjects. And for all datasets, reference images resemble each other *w.r.t.* pose and scale. As such

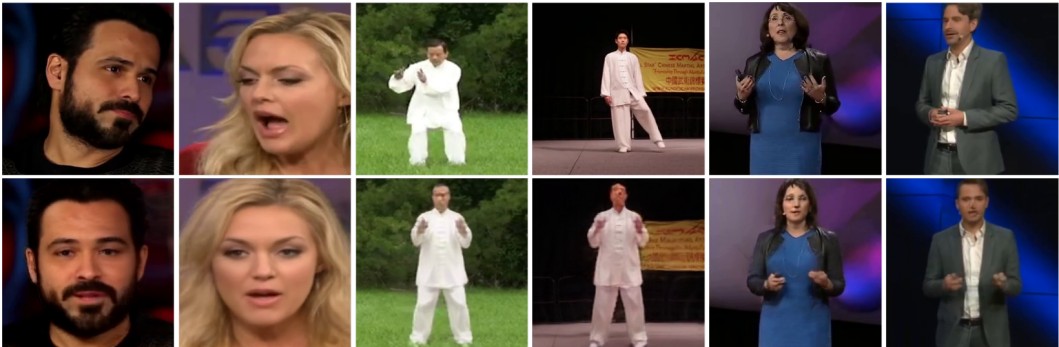

Figure 5: **Visualization of *reference images*.** Example source (top) and reference images (down) from VoxCeleb, TaichiHD and TED-talk datasets. Our network learns *reference images* of a consistently frontal pose, systematically for all input images of each dataset.

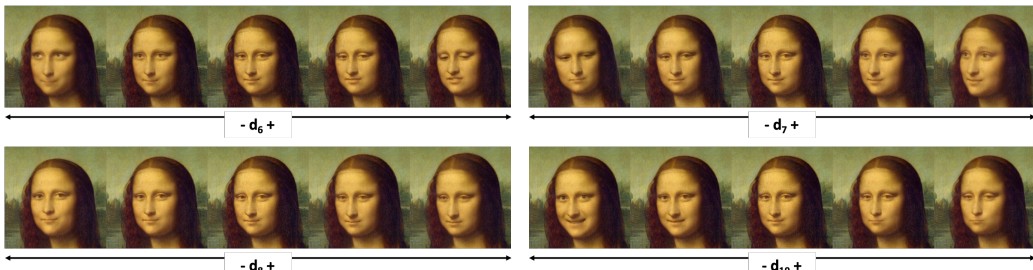

Figure 6: **Linear manipulation of four motion directions on the painting of Mona Lisa.** Manipulated results indicate that $d_6$ represents *eye movement*, $d_8$ represents *head nodding*, whereas $d_{19}$ and $d_7$ represent facial expressions.

reference images can be considered as a normalized form of $x_s$, learning transformations between $x_s$ and $x_d$ using $x_s \rightarrow x_r \rightarrow x_d$ is considerably more efficient than $x_s \rightarrow x_d$, once $x_r$ is fixed.

Noteworthy, we found the similar idea of learning a 'reference image' has also been explored by Siarohin et al. (2019) (FOMM) and Wiles et al. (2018) (X2Face). However, deviating from our *visualized* 'reference image', the 'reference image 'in FOMM refers to a non-visualized and abstract concept. In addition, LIA only requires a latent code $z_{s \rightarrow r}$, rather than the 'reference image' for both, training and testing, which is contrast to X2Face.

**(g) Motion dictionary interpretation.** Towards further interpretation of directions in $D_m$, we conduct linear manipulations on each $d_i$. Images pertained to manipulating four motion directions are depicted in Fig. 6. The results suggest that the directions in $D_m$ are semantically meaningful, as they represent basic visual transformations such as head nodding ($d_8$), eye movement ($d_6$) and facial expressions ($d_{19}$ and $d_7$). More results can be found on our project webpage[2].

## 5 CONCLUSIONS

In this paper, we presented a novel self-supervised autoencoder LIA, aimed at animating images via latent space navigation. By the proposed Linear Motion Decomposition (LMD), we were able to formulate the task of transferring motion from driving videos to source images as learning linear transformations in the latent space. We evaluated proposed method on real-world videos and demonstrated that our approach is able to successfully animate still images, while eliminating the necessity of *explicit structure representations*. In addition, we showed that the incorporated motion dictionary is interpretable and contains directions pertaining to basic visual transformations. Both quantitative and qualitative evaluations showed that LIA outperforms state-of-art algorithms on all benchmarks. We postulate that LIA opens a new door in design of interpretable generative models for video generation.

---

[2]https://wyhsirius.github.io/LIA-project/

## ETHIC STATEMENT

In this work, we aim to synthesize high-quality videos by transferring motion on still images. Our approach can be used for movie production, making video games, online education, generating synthetic data for other computer vision tasks, etc. We note that our framework mainly focuses on learning how to model motion distribution rather than directly model appearance, therefore it is not biased towards any specific gender, race, region, or social class. It works equally well irrespective of the difference in subjects.

## REPRODUCIBILITY STATEMENT

We assure that all the results shown in the paper and supplemental materials can be reproduced. We intend to open-source our code, as well as trained models.

## ACKNOWLEDGEMENTS

This work was granted access to the HPC resources of IDRIS under the allocation AD011011627R1. It was supported by the French Government, by the National Research Agency (ANR) under Grant ANR-18-CE92-0024, project RESPECT and through the 3IA Côte d'Azur Investments in the Future project managed by the National Research Agency (ANR) with the reference number ANR-19-P3IA-0002.

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

## A   DETAILS OF MODEL ARCHITECTURE

We proceed to describe the model architecture in this section. Fig. 7 shows details of our $E$. In each ResBlock in $E$, spatial size of input feature maps are downsampled. We take feature maps of spatial sizes from $8 \times 8$ to $256 \times 256$ as our appearance features $x_i^{enc}$. We use a 5-layer MLP to predict a magnitude vector $A_{d \to r}$ from $z_{d \to r}$. Fig. 8 (a) shows the general architecture of our $G$, which consists of two components, a flow field generator $G_f$ and a refinement network $G_r$. We apply *StyleConv* (Upsample + Conv3 × 3), which is proposed by StyleGAN2, in $G_f$. *StyleConv* takes latent representation $z_{s \to t}$ as style code and generates flow field $\phi_i$ and corresponding mask $m_i$. $G_r$ uses *UpConv* (Conv1 × 1 + Upsample) to upsample and refine inpainted feature maps to target resolution. We show details pertaining to $G$ block in Fig. 8 (b). Each $G$ block is used to upsample ×2 the previous resolution. We stack 6 blocks towards producing 256 resolution images.

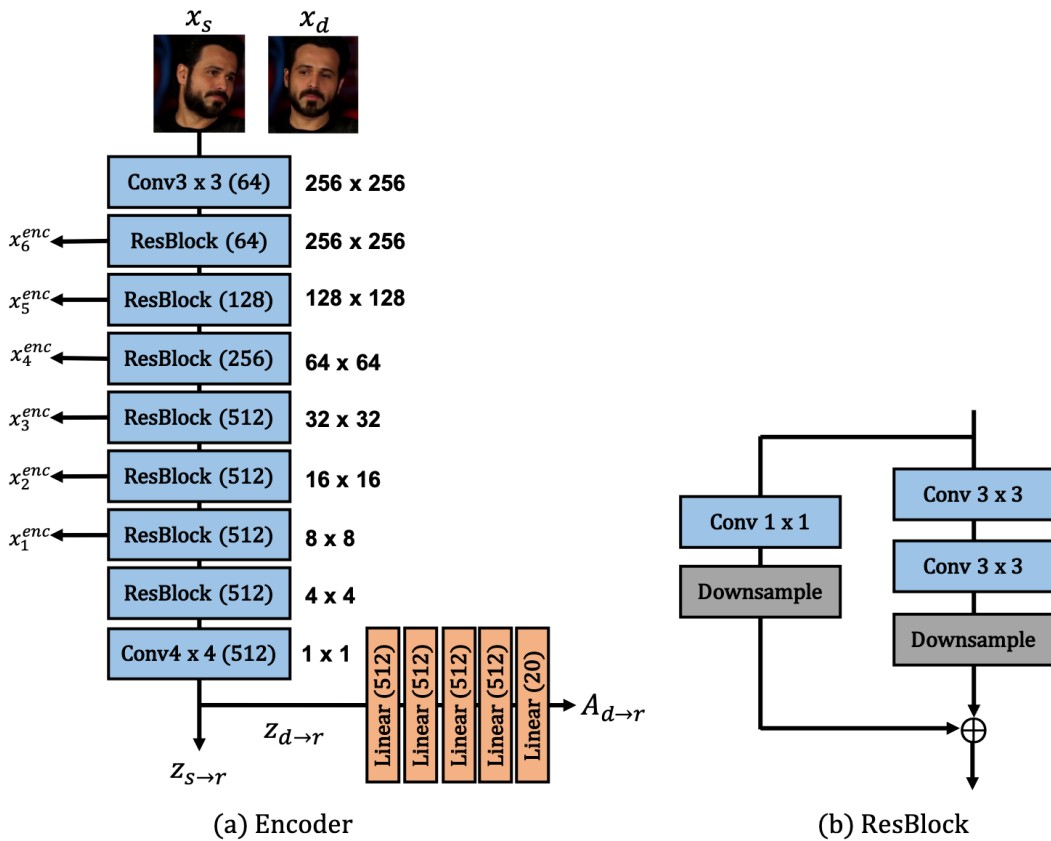

(a) Encoder                    (b) ResBlock

Figure 7: **Encoder architecture.** We show details of architecture of $E$ in (a) and ResBlock in (b).

## B   EXPERIMENTS

We proceed to introduce details of datasets and evaluation metrics used in our experiments.

### B.1   DATASETS

**VoxCeleb** (Nagrani et al., 2019) consists of a large amount of interview videos of different celebrities. Following the process of FOMM (Siarohin et al., 2019), we extract frames and crop them into $256 \times 256$ resolution. In total, VoxCeleb contains a training set of 17928 videos and a test set of 495 videos.

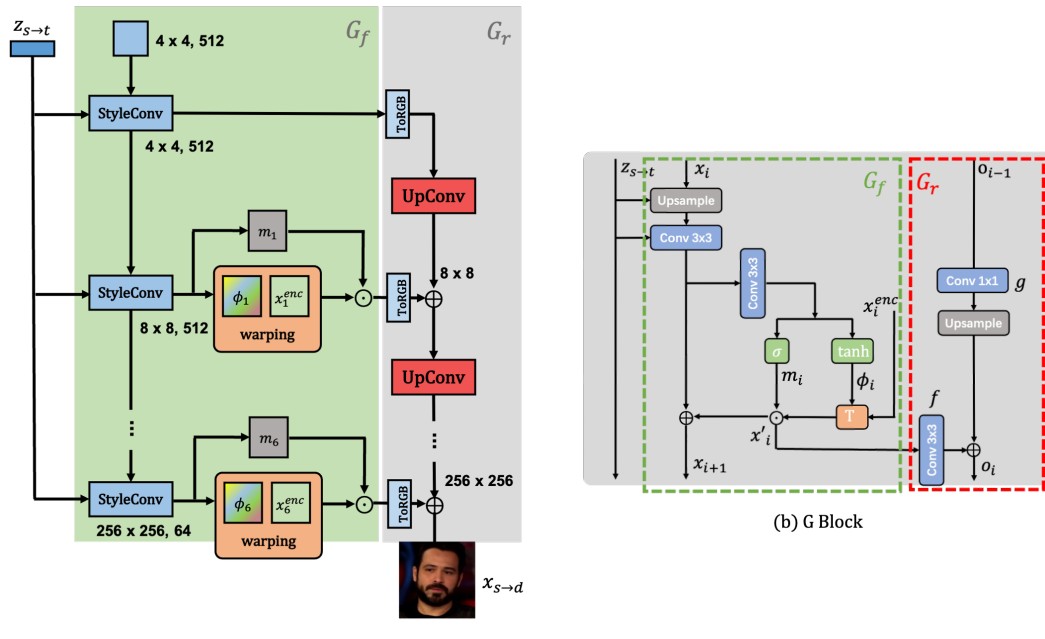

(a) Generator architecture

(b) G Block

Figure 8: **Generator architecture.** We show details about architecture of $G$ in (a) and $G$ block in (b).

**TaiChiHD** (Siarohin et al., 2019) consists of videos of full human bodies performing Tai Chi actions. We follow the original pre-processing of FOMM (Siarohin et al., 2019) and utilize its $256 \times 256$ version. TaiChiHD contains 1096 training videos and 115 testing videos.

**TED-talk** is a new dataset proposed in MRAA (Siarohin et al., 2021). It comprises a number of TED-talk videos, where the main subjects have been cropped out. We resize the original version into $256 \times 256$ resolution to train our model. This dataset includes 1124 training videos and 130 testing videos.

### B.2 EVALUATION METRICS

We use five different metrics to evaluate our experimental results, namely $\mathcal{L}_1$, LPIPS, AKD, MKR and AED that quantify the reconstructed results. In addition, we compute video FID to evaluate video quality in motion transferring tasks.

$\mathcal{L}_1$ represents the mean absolute pixel difference between reconstructed and real videos.

**LPIPS** (Zhang et al., 2018) aims at measuring the perceptual similarity between reconstructed and real images by leveraging the deep features from AlexNet (Krizhevsky et al., 2012).

**Video FID** is a modified version of the original FID (Heusel et al., 2017). We here follow the same implementation as Wang et al. (2020a) and utilize a pre-trained ResNext101 (Hara et al., 2018) to extract spatio-temporal features to compute the distance between real and generated videos distributions. We take the first 100 frames of each video as input of the feature-extractor to compute the final scores.

**Average keypoint distance (AKD) and missing keypoint rate (MKR)** evaluate the difference between keypoints of reconstructed and ground truth videos. We extract landmarks using the face alignment approach of (Bulat & Tzimiropoulos, 2017) and extract body poses for both TaiChiHD and TED-talks using OpenPose (Cao et al., 2019). AKD is computed as the average distance between corresponding keypoints, whereas MKR is the proportion of keypoints present in the ground-truth that are missing in a reconstructed video.

**Average Euclidean distance (AED)** measures the ability of preserving identity in reconstructed video. We use a person re-identification pretrained model (Zheng et al., 2020) for measuring human bodies (TaichiHD and TED-talk) and OpenFace (Amos et al., 2016) for faces to extract identity embeddings from reconstructed and ground truth frame pairs, then we compute MSE of their difference for all pairs.

### B.3 COMPARISON WITH FULL MRAA

We show quantitative evaluation results with the full MRAA model in Tab. 6. We observe that our method achieves competitive results in reconstruction and keypoint evaluation. While we do not explicitly predict keypoints, *w.r.t.* the TaichiHD dataset, interestingly we outperform MRAA in both, AKD and MKR. Such results showcase the effectiveness of our proposed method on modeling articulated human structures. However, reconstruction evaluation cannot provide a completely fair comparison on how well the main subjects (*e.g.,* faces and human bodies) are generated in videos. This is in particular the case for TaichiHD and TED-talk, where backgrounds have large contributions to the final scores.

Table 6: **Comparison with full MRAA.**

| | VoxCeleb | | | | TaichiHD | | | | TED-talks | | | |
|--------|-------|-------|-------|-------|-------|-------------------|-------|-------|-------|------------------|-------|-------|
| Method | $\mathcal{L}_1$ | AKD | AED | LPIPS | $\mathcal{L}_1$ | (AKD, MKR) | AED | LPIPS | $\mathcal{L}_1$ | (AKD, MKR) | AED | LPIPS |
| MRAA | **0.041** | **1.303** | **0.135** | 0.124 | **0.045** | (5.551, 0.025) | 0.431 | **0.178** | 0.027 | (**3.107**, **0.0093**) | **0.379** | **0.11** |
| Ours | **0.041** | 1.353 | 0.138 | **0.123** | 0.057 | (**4.823**, **0.020**) | **0.431** | 0.180 | **0.027** | (3.141, 0.0095) | 0.399 | **0.11** |

### B.4 REFERENCE IMAGE GENERATION.

To produce $x_r$, we use $G$ to decode $z_{s \to r}$ into the flow field $\phi_{s \to r}$. Reference image $x_r$ is obtained by warping $x_s$ using $\phi_{s \to r}$. The entire process is shown in Fig. 9.

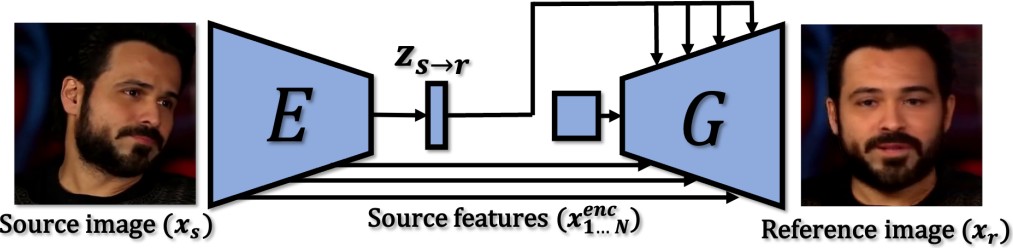

Figure 9: **Reference image generation.**

### B.5 QUALITATIVE RESULTS ON EFFECTIVENESS OF USING MOTION DICTIONARY

Fig. 10 illustrates the generated results on transferring motion from VoxCeleb to GermanPublicTV with and without motion dictionary. We observe that without the motion dictionary, appearance information is undesirably transferred from driving videos to generated videos.

### B.6 LIMITATIONS

For human body, one limitation of our method is dealing with body occlusion. We observe in Fig. 11 that in taichi videos, in case of occlusion cause by legs and arms, motion is not transferred successfully. In addition, in TED-talks, transferring hand motion is challenging, as hands are of small size, articulated and sometimes occluded by human bodies.

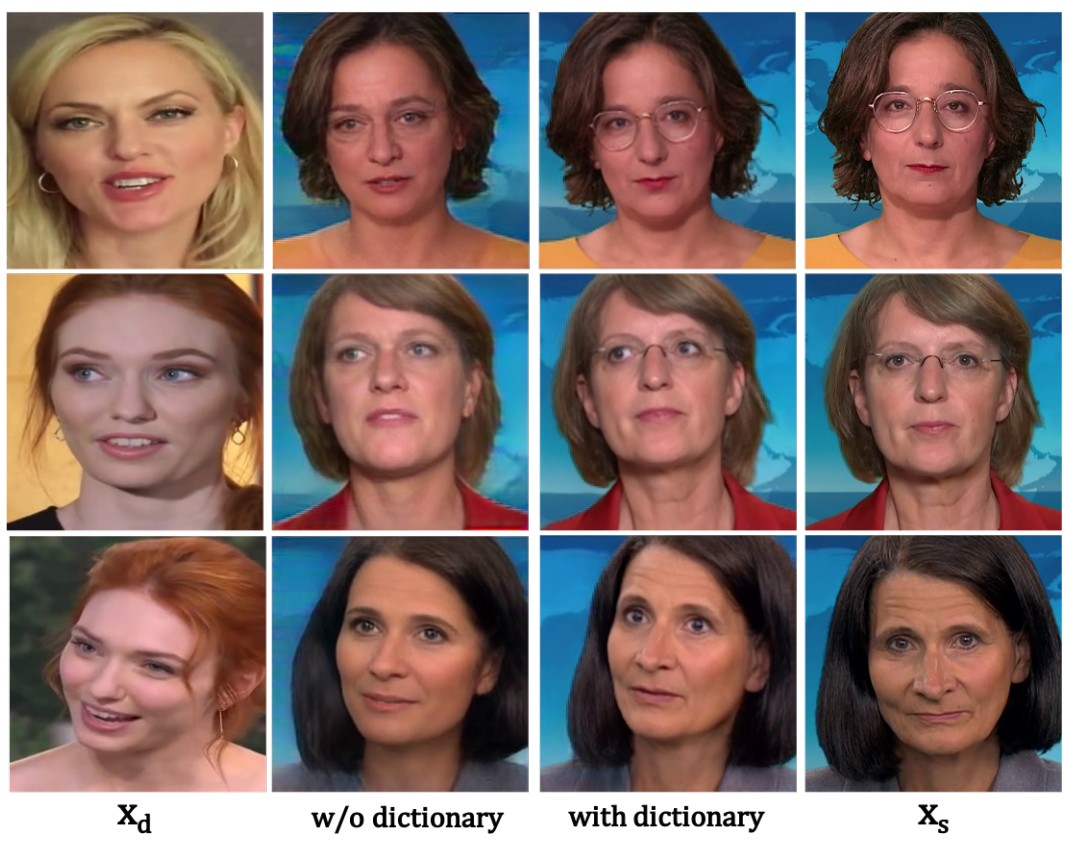

**$X_d$**         **w/o dictionary**         **with dictionary**         **$X_s$**

Figure 10: **Generated results with and without $D_m$.** We observe that the disentanglement of appearance and motion is much better by using $D_m$.

TaichiHD                         TED-talks

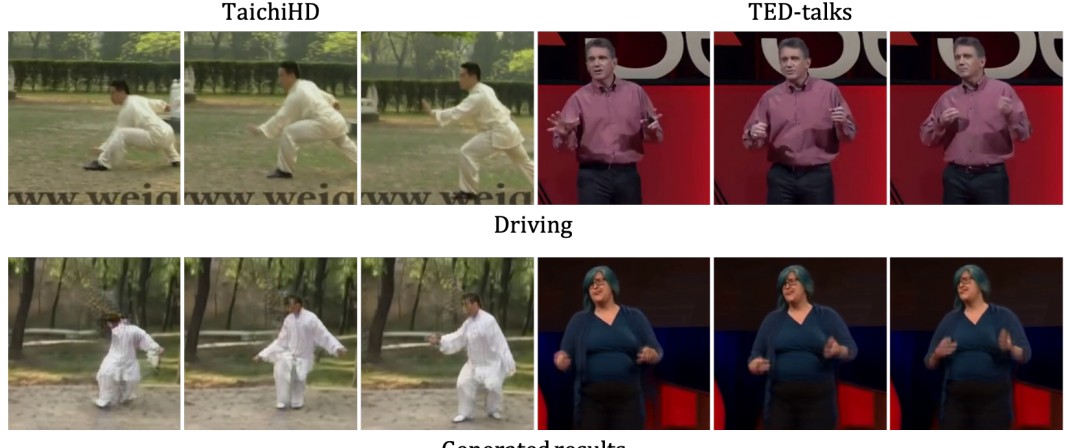

Driving

Generated results

Figure 11: **Failure cases.** We observed that it is still challenging for LIA to handle arm-leg occlusion (Taichi) and hand motion (TED-talk).

