# OpenReview forum: "Latent Image Animator: Learning to Animate Images via Latent Space Navigation"
_ICLR.cc/2022/Conference — ICLR 2022 Poster_

### Official Review · Reviewer_UntW · 2021-11-01

**Correctness:** 4
**Technical Novelty And Significance:** 3
**Empirical Novelty And Significance:** 3
**Recommendation:** 8
**Confidence:** 4

**Details Of Ethics Concerns:**

The paper proposes a completely implicit and end-to-end learned technique to animate subjects. As such this technique is very likely able to pick any data bias present in the training sets (e.g., under-represented people of color in face datasets). The authors do not mention this throughout the paper, I would suggest adding a section detailing possible concerns regarding the data bias present in completely data driven techniques.

Moreover, part of the experiments rely on a user study, however details on this study (besides the number of participants) have not been discussed in the paper. For example it is relevant for the reader to know if the participants are experts in the field, or random people, and also consider how they have been selected. This is to mitigate possible involuntary biases emerging from picking raters which are not representative of the final end-users.


**Main Review:**

Overall I found the paper quite interesting and the results are quite impressive. Some parts of the presentation could be improved. I’ll list below some of the strengths and weaknesses of the paper in the current form.

STRENGTHS

+ The proposed method is significantly more streamlined w.r.t. to the main competitor that relies on breaking the motion to animate into smaller local motions. The result is that the model itself as well as the training regime is significantly simpler to follow and (probably) to reimplement based on the information in the paper.

+ The quality of the results shown is impressive and the quantitative analysis confirms the improvement w.r.t. previously published methods.

WEAKNESSES

1. The proposed technique does not explicitly disentangle the motion of the foreground object from the motion of the background, while one of the contributions of the main competitor (i.e., MRAA) was handling exactly this. This can generate unwanted artifacts where the background seems to stick to the motion of the foreground subjects. This effect is clearly visible in the  `realworld_vox.mp4` video in the supplementary material.

2. The method presentation in section 3 could be improved.

     **a.** While reading, for me, it was not clear what $z_{s \leftarrow r}$ was supposed to represent till I read section 4.4. I would suggest moving part of this explanation directly in Section 3 in the form of intuitions about what the network it is supposed to model, and leave the experimental verification to Section 4.4.

    **b.** It’s not clear why the two images $x_s$ and $x_d$ should be processed by the same encoder with shared parameters. Per my understanding, the outcome of $E(x_d)$ would be a latent transformation $z_{d \rightarrow r}$ which the following MLP needs to “invert” and express as a combination of the learned basis vectors. Have the author considered using a different encoder for directly predicting $z_{r \rightarrow d}$?

3. The motion dictionary expressed as an orthogonal basis in the latent space is a pretty cool idea, but it could have been evaluated more deeply in the ablation study. In particular:

    **a.** Table 5 seems to suggest an increase in performance connected with an increase of dimensions in the learned dictionary, however the evaluation stops at 20. Is there anything preventing the authors from trying more than 20 dimensions?

    **b.** Tab 4 shows the impact of using a dictionary to model the motion, however there isn’t an ablation study testing whether the orthogonality constraint is needed and/or is beneficial to performance. I wonder what would happen if you express the motion as a linear combination of motions without constraints.

4. The fact that nothing is specifically designed in the network to force $z_{s \rightarrow r}$ to represent a transformation that brings the source image in a learned reference pose is both a strength and a weakness of the method as it may expose it to unexpected failures when applying it to a new set of data where such canonical pose cannot be clearly learned from the data.

5. Failure cases of the model are not discussed neither in the main paper nor in the supplementary material. This should be a good practice for any research work, but in particular for this model as moving from a more structured intermediate representation of the motion, to a completely implicit one could expose the model to very unexpected and catastrophic failures. I would encourage the authors to discuss some of these in the paper.

TYPOS & ERRORS

* Page 5 after equation 2: “Semantically each $d_i$ should represent an basic” -> “a basic”
* Beginning of Section 3.4.: “In reference” -> “At inference” ??
* First paragraph of section 4: (c) is mentioned two times, the first one might be a typo
* Is the driving image in the second row of Fig. 4 left correct? The poses of the generated images seem completely off.
* Tab 6 of the supplementary, please highlight with bold even numbers from the competitor.


**Summary Of The Paper:**

This work introduces a novel method for self-supervised image animation using a source image for the identity of the subject to animate (e.g., a face) and a driving video, involving another subject, showing the motion to replicate (e.g., another face speaking).
The main novelty of the work is to avoid explicit representation for the subject motion. The previous works decompose the subject into known parts (e.g., keypoints or regions) and defy local trajectories for them. Instead, the authors rely on an implicit representation of the motion as a trajectory in a learned latent space. The proposed architecture, given two frames, a source and a target one, generates an implicit transformation that brings the source one in a learned canonical reference pose and another transformation to go from canonical pose to target one. The two transformations combined can be decoded into the desired animation. This formulation greatly simplifies the network architecture and training regime w.r.t. previous works,without sacrificing the reconstruction quality. The proposed system achieves SOTA results in commonly used benchmarks for this field.


**Summary Of The Review:**

This work is quite interesting as it shows how it is possible to achieve compelling results in image animation without requiring any explicit intermediate structured representation for the motion of the main subject. The work builds on recent findings on the expressive power of the latent spaces of deep generative models for image manipulation and repurpose these for the task of animating images.
As a side effect of this drastically different approach, the method simplifies quite significantly the network architecture and the training regime while achieving equal or better results on standard benchmarks and user validation wrt the sota. The method presentation could be improved but the paper does not present major flaws in terms of methodology and experimental validation. For this reason I’m currently suggesting acceptance and rate  this paper at 8.

---

> ### Author Response · Authors · 2021-11-20
> **Response from the Authors**
>
> We thank the reviewer for the positive feedback and agree that 1) our idea is quite interesting, and 2) our real-world experiments are impressive.
>
> We have updated the manuscript to fix the typos and errors.
>
> **Q2.b. It’s not clear why the two images $x_s$ and $x_d$ should be processed by the same encoder with shared parameters. Have the author considered using a different encoder for directly predicting $w_{r\rightarrow d}$?**
>
> **A2.b.** Indeed, we have considered this and we found that directly predicting $w_{r\rightarrow d}$ is challenging. in the updated Supplementary Material. Without constraint in the latent space, the canonical pose cannot be learned very well, as the search space is substantially larger. Therefore, our idea is to learn a motion dictionary and predict the magnitude vector $A_{r\rightarrow d}$ from $z_{d\rightarrow r}$ to use a linearity constrain. Since $z_{s\rightarrow r}$ and $z_{d\rightarrow r}$ represent the same type of motion transformation (from input images to reference images), we consider that they should require the same type of structure features. Therefore, we process $x_s$ and $x_d$ using the same Encoder. In experiments, we also observed that a single encoder is able to learn a better canonical pose than two encoders, therefore we kept this setting.
>
> **Q3.a. Study on using more directions in motion dictionary.**
>
> **A3.a.** We have updated the manuscript and provided the quantitative results by exploring 40 and 100 directions on three datasets in Tab. 5. We illustrate qualitative results in $\color{brown}{comparison\underline{}md\underline{}size.mp4}$ in the updated Supplementary Material.
>
> To determine the size of the motion dictionary, we tested 5, 10, 20, 40 and 100 motion directions. In case of a dictionary size of 20, we observe that generated results are of satisfying visual quality, achieving superior or equal quantitative results compared to SOTA. As shown in $\color{brown}{comparison\underline{}md\underline{}size.mp4}$, models with 5 and 10 directions are limited w.r.t. modeling complex motion. For instance, some motion patterns are clearly missing in the generated results. In case of a motion dictionary endowed with 20 or more motion directions, animated results are of high quality, however the difference between models appears minor. Due to such observations, we select a motion dictionary comprising 20 motion directions.
>
> **Q3.b. I wonder what would happen if you express the motion as a linear combination of motions without orthogonality constraints.**
>
> **A3.b.** We provided comparison in Tab.7. We learn a non-orthogonal basis by randomly initializing 20 directions without using the Gram-Schmidt process. Tab.7 shows that models using orthogonal basis outperform the ones with a non-orthogonal basis. To further analyze the correlation between each direction after training, we compute angles between all direction-pairs in the motion dictionary for three datasets and show results in Fig.13, Fig.14 and Fig.15. We observe that most directions are nearly orthogonal after training, angles are within $90^{\circ} \pm 10^{\circ}$. Therefore, an orthogonal basis is effective on the three training datasets. However, there are other factors that may affect the such choice, e.g., complexity of the dataset and latent code dimension. It is an open question, whether orthogonality is a general solution or just in our context. We intend to investigate this in future work.
>
> **Q4. The method may fail when applied to a new set of data where such canonical pose cannot be clearly learned from the data.**
>
> **A4.** We agree with the reviewer. The goal of this paper is to conduct image animation on face and human body, which contain clear canonical poses. We will conduct experiments on more complex data such as horses and birds to investigate the generalizability of our method in future work.
>
> **Q5. Failure cases discussion.**
>
> **A5.** We uploaded $\color{brown}{failure\underline{}cases.mp4}$ in the updated Supplementary Material to showcase failure cases. For human body, one limitation of our method is dealing with body occlusion. The video shows two failure cases from our method, we can see that in TaichiHD, when there is occlusion caused by legs and arms, motion is not transferred successfully. In addition, we can observe that in TED-talks, transferring hand motion is challenging, as hands are small, articulated and sometimes occluded with human body.
>
> **Q6. Details on user study**
>
> **A6.** We note that the small-scale user study serves to verify the validity of the quantitative results. The 20 participants were random and anonymous researchers, willing to participate in the online user study.  We did not collect demographic data pertaining to the participants, who were non experts in the field of video generation. Noteworthy, the obtained human preference is in accordance with our quantitative evaluation. We include this description in the Supplementary Material.

---

> > ### Comment · Reviewer_UntW · 2021-11-26
> > **Thank you for the answers and final rating**
> >
> > I wish to thank the authors for addressing most of my questions and adding extra material to the paper that in my opinion made it stronger.
> > I rated this paper positively at 8 and after reading the other reviews and the reply from the authors I would like to keep the original rating and suggest acceptance.

---

### Official Review · Reviewer_gSih · 2021-11-02

**Correctness:** 4
**Technical Novelty And Significance:** 2
**Empirical Novelty And Significance:** 2
**Recommendation:** 6
**Confidence:** 4

**Main Review:**

+ves:
1. The idea of using learned linear orthogonal basis to replace representation to replace structure representations and re-use encoder is interesting.
2. Overall, the paper is clear, well written and easy to follow.
3. The comparison and ablation studies are intensive and well structured. The video for the real world case is impressive.

Concerns:
1. I am not fully convinced that the proposed linear orthogonal basis motion representation would be better than implicit structure landmark representations used in FOMM.  What is the dimension of this latent representation? And how to use it to generate dense flow? For a highly compact latent representation might not be easy to represent the complex motion and pose. In Fig. 4 row 2 and 3, the generated results’ pose seems less consistent with driving video. Also, the left-bottom video in ”comparison_taichi&ted.m4v”, the quality of video generated by MRAA seems better than the proposed method, especially for large poses of the face.

2. Since the encoder is shared for appearance and motion, how to ensure these two could be well disentangled? A deeper analysis of the proposed method would have been nice.

3. The main contribution of the proposed method is to use a learned linear motion dictionary, and can improve the generated image quality. On the other hand, would the linear assumption limit to generate large and complex motion?


**Summary Of The Paper:**

This paper proposes a self-supervised auto-encoder based pipeline for animating images via latent space navigation. To enable animation, they designed a Latent Image Animator (LIA) module to model motion using a learned linear orthogonal basis. Given an input source and target image, their pipeline first encodes images into latent space, further estimates an offset vector from the target image, and produces a dense flow field/ The final output image is generated by warping the source image and a multi-scale refinement network.
The main contribution of this paper is the introduction of a learned linear orthogonal basis motion representation to replace structure representations like landmarks. Besides this, they use a single encoder to extract appearance and motion information to reduce model size. Studies are performed to show the superiority of their proposed method over two SOTA approaches, including FOMM and MRAA on 3 datasets. Results are shown on the task of (1) source reconstruction from video. (2) motion transfer. (3). User study.


**Summary Of The Review:**

On balance I'm somewhat positive about the paper.  However, I have found some considerable issues as described above.

---

> ### Author Response · Authors · 2021-11-20
> **Response from the Authors (part 1/2)**
>
> We thank the reviewer for the positive feedback and agree that 1) our idea is interesting, 2) our paper is well-written, and 3) our real-world experiments are impressive.
>
> **Q1.1. I am not fully convinced that the proposed linear orthogonal basis motion representation would be better than implicit structure landmark representations used in FOMM. What is the dimension of this latent representation? And how to use it to generate dense flow?**
>
> **A1.1.** We have quantitatively and qualitatively demonstrated that our method outperforms FOMM. The dimension of each motion direction is $512$. The obtained latent codes are sent to each layer in our style-based generator, which produces multi-scale dense flows and corresponding occlusion masks (see Fig.7 and Fig.8 in the updated manuscript).
>
> There are two drawbacks of explicit structure representations. 1) Keypoints from FOMM are 2D- rather than 3D-aware representation. It is challenging to transfer 3D motion such as rotation and zooming from driving video using such representations. 2) Keypoints are sparse. Dense flow fields obtained from such sparse representations are not very precise and not able to preserve the structure of faces, if certain points are not well-predicted. As shown in  $\color{brown}{comparison\underline{}vox.mp4}$ and $\color{brown}{comparison\underline{}relative\underline{}absolute.mp4}$ in Supplementary Material, we can clearly observe that when transferring large motion, especially rotation, generated faces from FOMM and MRAA become deformed and lose their original shapes.
>
> Latent spaces of our model contain directions representing both 3D-aware and 2D-aware transformations, which can be seen from  $\color{brown}{vox256\underline{}linear\underline{}manipulation.mp4}$, $\color{brown}{vox512\underline{}linear\underline{}manipulation.mp4}$, $\color{brown}{taichi256\underline{}linear\underline{}manipulation.mp4}$ and $\color{brown}{ted256\underline{}linear\underline{}manipulation.mp4}$ in the updated Supplementary Material.
>
> For example for VoxCeleb in $\color{brown}{vox256\underline{}linear\underline{}manipulation.mp4}$, $d_0$, $d_5$, $d_7$, $d_8$ and $d_{18}$ represent a set of 3D motion such as rotation, zooming and shifting, and the remaining directions correspond to facial dynamics such as eye and mouth movement. In $\color{brown}{vox512\underline{}linear\underline{}manipulation.mp4}$, $d_1$, $d_3$, $d_{10}$ and $d_{13}$ represent 3D motion, especially $d_{13}$, which clearly shows the rotation of the head, and the rest directions indicate facial dynamics.
>
> **Q1.2. For a highly compact latent representation might not be easy to represent the complex motion and pose. In Fig. 4 row 2 and 3, the generated results’ pose seems less consistent with driving video.**
>
> **A1.2.** In our real-world experiments $\color{brown}{realworld\underline{}vox.mp4}$ and $\color{brown}{realworld\underline{}vox512.mp4}$, we showcase that highly compact latent representations are very powerful and sufficient to transfer complex poses and motions even in high resolution ($512\times 512$).
>
> Our method is able to replicate motion well. Fig. 4 might be misleading, as we illustrate the results using relative instead of absolute motion transfer mode, without displaying the first frame of the driving video (discussed in Section 3.4). In contrast, absolute motion transfer generally imposes the head pose of the driving video onto the source image. Descriptions have been added in the revised manuscript
>
> Existing methods including ours, FOMM and MRAA support both, absolute and relative motion transfer modes. In the updated Supplementary Material, $\color{brown}{comparison\underline{}relative\underline{}absolute.mp4}$ is the video version of Fig. 4, where we show generated videos from different methods using both modes.
>
> We note that in **relative transfer mode**, as also mentioned in FOMM, source image and the first frame of a driving video are required to encompass a similar starting facial pose. In case that two images have large pose variations, the motion of a face in a driving video can be well transferred, however will start from the pose of source image. Since in our cross-dataset experiment, image-video pairs between FFHQ and VoxCeleb are randomly constructed, we do not have a pose-similarity constraint for each pair. Hence pairs, as shown in Fig. 4, might contain similar (last row) or substantially different poses (second row).
>
> As shown in $\color{brown}{comparison\underline{}relative\underline{}absolute.mp4}$, facial appearance is well-preserved in relative transfer mode, whereas absolute transfer might create artifacts, especially when source and driving images have large pose discrepancy. However, we observe that our method outperforms the SOTA also in absolute transfer mode.

---

> > ### Author Response · Authors · 2021-11-20
> > **Response from the Authors (part 2/2)**
> >
> > **Q1.3. The left-bottom video in ${comparison\underline{}taichi\underline{}ted.mp4}$, the quality of video generated by MRAA seems better than the proposed method, especially for large poses of the face.**
> >
> > **A1.3.** We respectfully disagree, as we do not observe  this in $\color{brown}{comparison\underline{}taichi\underline{}ted.mp4}$. In addition,  we direct the reviewer to the quantitative evaluation in Tab. 1, which reports method outperforms MRAA w.r.t.  reconstruction on the TED-talk dataset
> >
> > **Q2. Since the encoder is shared for appearance and motion, how to ensure these two could be well disentangled? A deeper analysis of the proposed method would have been nice.**
> >
> > **A2.** Our shared Encoder has two outputs, (a) latent codes $z_{s\rightarrow r}$\/$z_{d\rightarrow r}$
> > and (b) multi-scale feature maps $x_{s}^{enc}$. We use $x_{s}^{enc}$ to represent appearance, and $z_{s\rightarrow r}$/$z_{d\rightarrow r}$ to represent motion transformations from $x_s$/$x_d$ to the reference image $x_r$. We use a shared Encoder because $z_{s\rightarrow r}$/$z_{d\rightarrow r}$ also rely on structure features in $x^{enc}_{s}$.
> >
> > The key to disentangle motion in our framework is the motion dictionary $D_m$. It is an individual module and is learned independently. Since input images have no direct access to $D_m$, leaking of appearance from the driving image is able to be prevented, and hence $D_m$ is able to model motion transformations required for reconstruction objective during training.
> >
> > We have provided four videos, $\color{brown}{vox256\underline{}linear\underline{}manipulation.mp4}$, $\color{brown}{ted256\underline{}linear\underline{}manipulation.mp4}$, $\color{brown}{taichi256\underline{}linear\underline{}manipulation.mp4}$ and $\color{brown}{vox512\underline{}linear\underline{}manipulation.mp4}$ to visualize linear manipulation on motion dictionaries. From the videos, we can clearly observe that $D_m$ represent motion transformations.
> >
> > **Q3. Would the linear assumption limit to generate large and complex motion?**
> >
> > **A3.** In the latent space, our objective is to move the source representation to target representation, which can be achieved by both linear and non-linear navigation. While one single linear function might not be as powerful as a non-linear function, we jointly employ a set of linear directions, which model complex motion very well. We demonstrate by the ablation study related to the size of the motion dictionary that 20 directions are able to model motion distribution for current datasets.
> >
> > The advantage of using linear assumption is that navigation of each direction can be conducted very easily, and combination of multiple directions does not require extra operations. In the non-linear setting, modeling one potential transformation requires to recursively use an extra function $f$, which corresponds to a relatively small moving step in latent space, in order to achieve a large non-linear walk. Such design might increase the complexity of the whole framework, in particular when multiple $f$ are required. However, designing such non-linear functions require extensive experiments for trail and error of model architecture, which we envision to do in future work.

---

> > > ### Comment · Reviewer_gSih · 2021-11-29
> > > **Thank you for your answer**
> > >
> > > I am happy to see more results and new fig.4. The authors have clarified my questions about the pose mismatching issue. The new videos and figs are more clear than the original ones, especially for comparison. Thank you for providing those new results.
> > >
> > > Though this paper has quantitatively and qualitatively demonstrated outperforms than previous methods, I agree with Reviewer PbBc and jq2i about the limited novelty about unsupervised disentanglement of motion and appearance as well as implicit structural representation. Thus, I would like to keep my original rating as 6.

---

### Official Review · Reviewer_PbBc · 2021-11-03

**Correctness:** 4
**Technical Novelty And Significance:** 2
**Empirical Novelty And Significance:** 3
**Recommendation:** 6
**Confidence:** 4

**Main Review:**

Strengths:

 - The idea is interesting and well motivated based on recent literature on interpretable direction in the latent space of generative models.

 - The experimental results highlight the improvement over various methods using a variety of metrics. I would be interested to see comparison with different approaches for face re-enactment, for example audio-driven approaches (e.g., [1])

 - The paper is generally well-written.

Weaknesses:

 - The main weakness is the limited novelty of the method, since similar approaches have been utilised before, albeit not in the exact same way. For example, linear displacement of the latent codes using a motion basis is also used in [2] for video generation (not re-enactment). Similarly, a pre-trained encoder and generator are used in [3], where motion is synthesised again by shifting the latent code. However, I believe that the significance of the paper lies beyond the novelty of the method itself. Therefore, I would like to see some experimental comparison with some of these works, as well as discussion regarding the differences and advantages of the proposed method.

 - It would be helpful to see some qualitative results containing failure cases.

 - The impact of the orthogonality constraint on D is not studied. It would be interesting to see how it relates to the number of directions.


[1] End-to-end speech-driven facial animation with temporal gans

[2] A good image generator is what you need for high-resolution video synthesis

[3] Style Generator Inversion for Image Enhancement and Animation



**Summary Of The Paper:**

The paper proposes a method to animate images using a guide video. The novelty of the method lies in the absence of any form of structural representation. The authors pose the problem of synthesising motion as that of traversing the latent space. To this end,  a motion basis is learnt and is utilised to linearly shift each latent code. The experiments showcase the efficacy of the proposed model on a number of datasets.


**Summary Of The Review:**

Overall, this is an interesting paper with significant results. I lean towards accepting this work if the issues that were raised are resolved.

---

> ### Author Response · Authors · 2021-11-20
> **Response from the Authors (part 1/2)**
>
> We thank the reviewer for the positive feedback, stating that (1) our proposed idea is interesting and well motivated, that (2) experiments showcase the efficacy of our method, as well as that (3) the paper is well-written.
>
> **Q1. I would be interested to see comparison with different approaches for face re-enactment, for example audio-driven approaches.**
>
> **A1.** We agree that a comparison of image animation approaches based on different input modalities (e.g., video, audio and text) would be an intriguing study. However, it requires re-running our method on several different datasets for fair comparisons. We are in the process of conducting the experiments, but we are not certain to provide results during rebuttal time, we intend leave this studyin our future work
>
> **Q2. Novelty and comparison with [2] and [3]**
>
> **A2.** In [2], [3] and our work, the idea of latent code navigation is inspired from studies in GAN-interpretability, however it is not the main contribution in [2], [3], nor in our work. Deviating from [2] and [3], the novelty in our work has to do with disentangling the latent code. Our proposed Linear Motion Decomposition is able to learn interpretable and advanced disentangled motion representations for image animation.
>
> Our work differs from [2] twofold. (i) While
> [2] is mainly focused on generating videos from random noise (noise2video generation), our objective is motion transfer (image2video generation). The challenge in [2] has to do with effectively mapping a sequence of random noises into a spatio-temporal consistent videos, whereas our aim is to learn disentangled appearance and motion representations from input videos. (ii) Further, [2] obtains their basis inspired by [Shen et al., 2021], where they sample a large set of $z$, with a PCA being conducted to obtain a basis for latent code displacement. However, since the basis is fixed during training, we postulate that it is not able to obtain a well-disentangled motion space. Many directions correspond to **appearance** in such a space, as pointed out in [Shen et al., 2021]. We show a comparison with [2] in $\color{brown}{comparison\underline{}mocoganhd.mp4}$, where our method is able to better preserve appearance in generated videos.
>
> Compared to our work, there are three limitations of how [3] obtains identity and pose representations. (i) Due to lack of loss or pre-text task, constraining motion transformation, the obtained motion code is imprecise. Minor movement is likely to be ignored. For instance, $\color{brown}{comparison\underline{}inversion.png}$  in the updated Supplementary Material demonstrates that [3] ignores eye motion stemming from the driving image, whereas our method is able to re-target such fine motion. (ii) Without employing a basis, appearance information is able to leak into the motion code, which will lead to unsuccessful motion transfer. (iii) Using inversion to obtain code requires optimization for each image, which is time consuming and not very practical in real-world applications. Since the authors do not provide testing code for face animation, a fair quantitative and qualitative comparison might be hard to conduct.
>
> Our method differs from [2] and [3] in following three points. (i) We propose to decompose target motion representation $w_{r\rightarrow d}$ into a magnitude vector $A_{r\rightarrow d}$ and a motion dictionary $D_m$. $D_m$ is learned independently, where input images have no direct access. Such design can prevent the leaking of appearance. The ablation study in Sec. 4.3(d) proves the effectiveness of such design for successful motion transfer. (ii) Since $D_m$ is learnable, it is able to fit to motion distribution of the training dataset, which is different from [2]. (iii) We use QR decomposition, whereas PCA is used in [2]. We found QR is more efficient, providing a more uniform basis. While we only have 20 directions in our framework, [2] employs 384 directions. The videos $\color{brown}{vox256\underline{}linear\underline{}manipulation.mp4}$, $\color{brown}{ted256\underline{}linear\underline{}manipulation.mp4}$, $\color{brown}{taichi256\underline{}linear\underline{}manipulation.mp4}$ and $\color{brown}{vox512\underline{}linear\underline{}manipulation.mp4}$ show that $D_m$ contains various different motion transformations.
>
> **Q3. Failure cases.**
>
> **A3.** We uploaded $\color{brown}{failure\underline{}cases.mp4}$ in the updated Supplementary Material to show several failure cases and limitations of our method. In particular, when the body is occluded in the Taichi videos, motion of legs and arms is not transferred successfully. In TED-talks, the biggest challenge is to transfer hand motion, as hands are small and frequently occluded by the human bodies.

---

> > ### Author Response · Authors · 2021-11-20
> > **Response from the Authors (part 2/2)**
> >
> > **Q4. The impact of the orthogonality constraint on $D_m$ is not studied. It would be interesting to see how it relates to the number of directions.**
> >
> > **A4.** We provide a comparison in Tab. 7 in the updated manuscript. We learn a non-orthogonal basis by randomly initializing 20 directions without using the Gram-Schmidt process. Tab. 7 shows that models using orthogonal basis outperform the ones with a non-orthogonal basis. To further analyze the correlation between each direction after training, we compute angles between all direction-pairs in the motion dictionary for three datasets and show results in Fig.13, Fig.14 and Fig.15. We observe that most directions are nearly orthogonal after training, angles are within $90^{\circ} \pm 10^{\circ}$. Therefore, an orthogonal basis is effective on the three training datasets. However, there are other factors that may affect such choice, e.g., complexity of the dataset and latent code dimension. It is an open question, whether orthogonality is a general solution or just in our context. We intend to investigate this in future work.
> >
> > We have updated the manuscript by providing quantitative results on 40 and 100 directions on the three datasets in Tab.5. In addition, qualitative results are shown in $\color{brown}{comparison\underline{}md\underline{}size.mp4}$ in the updated Supplementary Material.
> >
> > To determine the size of the motion dictionary, we tested 5, 10, 20, 40 and 100 motion directions. In case of a dictionary size of 20, we observe that generated results are of satisfying visual quality, achieving superior or equal quantitative results compared to SOTA. As shown in $\color{brown}{comparison\underline{}md\underline{}size.mp4}$, models with 5 and 10 directions are limited w.r.t. modeling complex motion. For instance, some motion patterns are clearly missing in the generated results. In the case of a motion dictionary endowed with 20 or more motion directions, animated results are of high quality, however the difference between models appears minor. Due to such observations, we select a motion dictionary comprising 20 motion directions.

---

### Official Review · Reviewer_jq2i · 2021-11-04

**Correctness:** 3
**Technical Novelty And Significance:** 3
**Empirical Novelty And Significance:** 2
**Recommendation:** 6
**Confidence:** 4

**Main Review:**

Strengths:
+ The paper is well-written and insights of this work are clearly illustrated.
+ This work proposes to directly manipulate the latent space without need of explicit structure representation, such as keypoints or regions. The framework is elegant, relatively novel, and technically sound.
+ Qualitative and quantitative experiments demonstrate the superiority of the proposed method compared with previous arts. High-fidelity images (512x512 and 256x256)are shown.

Weaknesses:
- Though technically sound, the disentanglement of motion and appearance in an unsupervised manner is not entirely new, e.g.
MoCoGAN: Decomposing Motion and Content for Video Generation, CVPR 2018
Motion-Based Generator Model: Unsupervised Disentanglement of Appearance, Trackable and Intrackable Motions in Dynamic Patterns, AAAI 2020
- Some implementation details are missing, e.g. model architectures, latent code dimensions, etc.
- In terms of qualitative experiments, a sequence of generated images is expected. The model should not only produce realistic images  that match motion flow of the driving image, but should also maintain the temporal consistency within the driving video. Such performance should also be demonstrated.
- Visualization of some basic transformation $d_i$ or the generated optical flow can also be beneficial in showing the effectiveness of the motion dictionary.

**Summary Of The Paper:**

This work proposes LIA, a self-supervised AE that animates images by linear navigation in the latent space. LIA employs a two-step training paradigm. The first step encodes a source image into a latent code and then navigates to the target latent code along with a linear path. The second step decodes the target code to dense optical flow and warp with the source image. The framework is trained in a self-supervised manner. Experiments demonstrate the effectiveness of the proposed method on high-fidelity datasets.

**Summary Of The Review:**

This work proposes a relatively novel method that aims to animate still image via latent space navigation. Experiments show the state-of-the-art performance compared with previous works. This work, nevertheless, can further be improved by providing more supporting experiments.

---

> ### Author Response · Authors · 2021-11-20
> **Response from the Authors**
>
> We thank the reviewer for the positive feedback, in particular that (1) our paper is well-written and insights are clearly illustrated, that we (2) propose an elegant, relatively novel and technically sound approach, with which we are able to (3) synthesize high-fidelity (512x512 and 256x256) generated results.
>
> We have updated the manuscript to provide details of model architecture and latent code dimensions as suggested by reviewer.
>
> **Q1. Though technically sound, the disentanglement of motion and appearance in an unsupervised manner is not entirely new.**
>
> **A1.** Unsupervised disentangled representation learning is an unsolved problem for video understanding, as well as for video synthesis. The novelty in our work lies in the way we disentangle representations rather than the problem itself. Our contributions include (i) a simple Autoencoder architecture, which can disentangle appearance and motion by employing only one Encoder; (ii) a novel Linear Motion Decomposition module, which learns an interpretable linear basis as motion dictionary for transferring motion without explicit structure representations. MoCoGAN and [Xie et al. 2020] do not consider motion decomposetion, and only use a single vector to represent motion. Our model incorporates an advanced method for disentangling the latent space as opposed to MoCoGAN and [Xie et al. 2020], as we consider to represent motion space by using a linear combination of vectors from a learned motion basis, which is more interpreatble and better-disentangled. Generated results showcase the powerful animation capacity of our approach, specifically in transferring motion on high-resolution talking heads.
>
> **Q2. Temporal consistency demonstration?**
>
> **A2.** We provide $\color{brown}{realworld\underline{}vox.mp4}$, $\color{brown}{realworld\underline{}vox512.mp4}$, $\color{brown}{comparison\underline{}vox.mp4}$ and $\color{brown}{comparison\underline{}taichi\underline{}ted.mp4}$ in the updated Supplementary Material, which are aimed to demonstrate spatio-temporal consistency in our generated videos. In addition, we compute video FID, as well as AED to quantitatively evaluate temporal consistency following the protocols in MRAA. We note that video FID computes the distance between generated and real video distributions. We report the results in Tab. 2 in the manuscript. AED measures the identity consistency for both face and human body, which is reported in Tab. 1 of the manuscript.
>
> **Q3. Can the effectiveness of motion dictionary be demonstrated by visualization?**
>
> **A3.** We thank the reviewer for pointing this out. Visualization of motion dictionaries has been included in the updated Supplementary Material.
>
> In $\color{brown}{vox256\underline{}linear\underline{}manipulation.mp4}$, $\color{brown}{ted256\underline{}linear\underline{}manipulation.mp4}$, $\color{brown}{taichi256\underline{}linear\underline{}manipulation.mp4}$ and $\color{brown}{vox512\underline{}linear\underline{}manipulation.mp4}$, we show linear manipulation of each $d_i$ for VoxCeleb (256 resolution), TED-talks, TaichiHD and VoxCeleb (512 resolution), respectively. The results suggest that the motion dictionaries contain various motion transformations.

---

> > ### Comment · Reviewer_jq2i · 2021-11-29
> > **Re: Response from the Authors**
> >
> > Thank authors for clarification and additional supplementary materials. However, I didn't see any revision regarding the aforementioned references via comparisons or related work mentions. Please address this issue carefully. More references are listed here:
> > - Unsupervised learning of disentangled representations from video
> > - Disentangled sequential autoencoder
> > - Transmomo: Invariance-driven unsupervised video motion retargeting
> > - Disentangling multiple features in video sequences using gaussian processes in variational autoencoders

---

> > > ### Author Response · Authors · 2021-11-30
> > > **Re:Re:Response from the Authors**
> > >
> > > We thank the reviewer for listing references to interesting related works. We will include a discussion and comparison in the final version of our related work section.
> > >
> > > The named works[1][2][4] focus on learning disentangled representations in video sequences via different constraints. Denton and Birodkar [1] propose adversarial loss, which penalizes content towards learning pose features. Both content and pose features are then concatenated to synthesize images. Li and Mandt [2] introduce a probabilistic graphical model to learn dynamic, as well as content representations, respectively from videos/audios. Bhagat et al. [4] proposes a VAE, which uses Gaussian processes to model the latent space for the unsupervised learning of disentangled representations in video sequences.
> > >
> > > Our work differs from [1][2][4] in threefold. (1) *Task*. While the works [1][4] focus on video prediction and [3] shows its effectiveness in content swapping, we aim to **animate a single source image by using learned motion representations from a driving video**, which is not addressed in any of the listed works. (2) *Latent space constraint*. We propose a Linear Motion Decomposition module to learn a disentangled motion space towards generating target flow fields, which are used to warp appearance features to obtain final results. Deviating from [1][2][4], which directly predict motion (dynamic, pose) features, we use a learned basis to represent each motion code. This design allows us to better disentangle motion representations, as well as allows for interpretability. (3) *Generated quality*. Our work is able to synthesize 512 resolution complex human videos, whereas the works [1][2][4] are only able to produce low resolution and simple datasets (e.g., Moving MNIST and Sprites). In addition, our work uses flow fields to warp image features to obtain generated results while [1][2][4] predict all pixels from scratch. The works [1][2][4] are challenged in preserving the appearance information, in particular in real-world problem setting (e.g., zero-shot video generation).
> > >
> > > Yang et al. [3] mainly focus on generating pose sequences, which are then sent to a pre-trained renderer to conduct pose2vid generation. The work [3] differs from our work in twofold. (1) *Inputs*. While Yang et al. [3] leverages explicit structure poses as motion guidance, we do not. (2) *Zero-shot generation*. The work [3] requires training different rendering models for different subjects, which is not the case in our work. Once trained on face datasets, our model can be applied on any unseen faces without fine-tuning or re-training. In addition, we only require one image of a source subject, whereas in [3] videos of both source and target subjects are necessitated.
> > >
> > > [1] Unsupervised learning of disentangled representations from video. \
> > > [2] Disentangled sequential autoencoder. \
> > > [3] Transmomo: Invariance-driven unsupervised video motion retargeting. \
> > > [4] Disentangling multiple features in video sequences using gaussian processes in variational autoencoders.

---

> > > > ### Comment · Reviewer_jq2i · 2021-12-08
> > > > **Re: Re:Re:Response from the Authors**
> > > >
> > > > I would like to thank the authors for addressing most of my concerns. I would recommend this work for acceptance if all the aforementioned comments will be revealed in the final version. Additional missing references:
> > > >
> > > > [5] Motion-Based Generator Model: Unsupervised Disentanglement of Appearance, Trackable and Intrackable Motions in Dynamic Patterns

---

### Official Review · Reviewer_EXdL · 2021-11-05

**Correctness:** 4
**Technical Novelty And Significance:** 3
**Empirical Novelty And Significance:** 3
**Recommendation:** 8
**Confidence:** 3

**Details Of Ethics Concerns:**

The paper doesn't have specific ethic concerns to me.

**Main Review:**

I think the paper presented a good idea for learning image motion representation. The motion itself could be tricky to represent, especially within the image space for animation purposes. Previously used concepts such as key points often have limitations to sufficiently disentangle motion from other attributes such as shape. Therefore it is desirable to have a more interpretable and disentangled representation.

Strength:
1. The idea is novel to me. The motivation is clear that the paper wants to get rid of explicit structural representation for motion. The linear motion decomposition module is interesting and novel.

2. Experimental results are generally supportive. Compared to existing work, results seem to have less distortion and artifacts. The quantitative results also suggest the proposed method outperforms previous art.

I have some concerns about the paper as follows:

1. As explicit structural representation has its limitations. It also has the advantages of being very interpretable and can be learned in an unsupervised manner. The interpretability of the proposed method is not very clear to me. For example, how interpretable is the motion dictionary? The paper selectively showed some dimensions of the motion dictionary in Fig 6 and Appendix B.5. I think it might also be beneficial to show all of them. Do they also exhibit physical disentanglement such as 3D pose, expression? What about for human body?

2. The size of the dictionary is decided empirically, as described in 4.3. It is said to be empirically decided between 5, 10, and 20, where 20 yields the best reconstruction. I wonder why not go higher to 40 or even 100? Meanwhile, how does M value affect the learned motion dictionary in terms of disentanglement and interpretability?

3. In Figure 4, the proposed method doesn't seem to be able to replicate the driving video's pose very well. The paper should probably also compare with [Wang et al 2021a], which has an online demo.

4. Some concepts are not clarified the first time its mentioned, leading to confusion. For example, what is "r" in "Zs->r" is not explained until the experiments. In the generator, T is also not explained.

5. Some Typos exist in the paper. For example,  Section 3.4 , first paragraph, "In reference" --> "In inference".

**Summary Of The Paper:**

This paper presented a method for animate images of a certain category, e.g., faces or human bodies. Different from some previous work with explicit structural representations, this paper focuses on learning a set of linear latent space directions that controls image animation without explicit structural representations. The network learns a set of orthogonal linear basis of latent codes that can be linearly combined into a latent code that represents a motion from a "reference." The paper conducted a re-animation experiment on the human face and body motion dataset. Results are state-of-the-art by qualitative and quantitative measures.

**Summary Of The Review:**

I think the paper presented an interesting idea for learning latent motion representation. The results are also very encouraging. It would be great if there are more in-depth analysis about the learned representation in terms of interpretability and disentanglement. Some writing improvement would also help the readers to understand the details better.

---

> ### Author Response · Authors · 2021-11-20
> **Response from the Authors (part 1/2)**
>
> We thank the reviewer for the positive feedback, and that they have appreciated that (1) our proposed Linear Motion Decomposition is a novel and interesting idea for image animation, and (2) our qualitative and quantitative experiments are supportive.
>
> We have fixed typos and have provided a clarification of $x_r$ and $\mathcal{T}$ in the updated manuscript.
>
> **Q1. How interpretable is the motion dictionary? Do they also exhibit physical disentanglement such as 3D pose, expression? What about for human body?**
>
> **A1.** In the updated Supplementary Material, we have provided 4 videos to visualize linear manipulation results of each direction in the motion dictionaries learnt for four datasets. The results suggest that motion dictionaries in our models contain elements for both, 3D head poses and expressions for faces, as well as for the whole human body.
>
> In $\color{brown}{vox256\underline{}linear\underline{}manipulation.mp4}$, $d_0$, $d_5$, $d_7$, $d_8$ and $d_{18}$ represent 3D motion such as rotation, zooming and shifting, whereas the remaining directions correspond to different facial dynamics with various eye and mouth movements.
>
> In $\color{brown}{vox512\underline{}linear\underline{}manipulation.mp4}$, $d_1$, $d_3$, $d_{10}$, $d_{13}$ and $d_{16}$ represent 3D motion. In particular, $d_{13}$ and $d_{16}$ clearly indicate the rotation and zooming of head, respectively. The remaining directions indicate various facial dynamics.
>
> In $\color{brown}{ted256\underline{}linear\underline{}manipulation.mp4}$, we illustrate that $d_0$ and $d_{10}$ represent 3D rotation of human body, $d_7$, $d_{12}$ and $d_{19}$ represent head movements and the remaining directions correspond to joint hand and head movements. At the same time in TED-talks videos, presenters prefer to move their hands and heads simultaneously, hence motion of hand and heads are generally speaking entangled.
>
> TaichiHD is the most challenging dataset, as it contains complex motions with occlusions of arms and bodies. In $\color{brown}{taichi256\underline{}linear\underline{}manipulation.mp4}$, we notice that the interpretability of each direction is not very obvious, most of the directions represent joint motion between arms and legs due to the simultaneous movements of both body parts in Taichi videos.
>
> **Q2.1. The size of the dictionary is decided empirically, as described in 4.3. It is said to be empirically decided between 5, 10, and 20, where 20 yields the best reconstruction. I wonder why not go higher to 40 or even 100?**
>
> **A2.1.** We tested larger sizes of the motion dictionary and found that both, qualitative and quantitative results are not impacted by a larger dictionary.
>
> We have updated the manuscript and have provided quantitative results related to 40 and 100 motion directions on the three datasets (see Tab. 5). In addition, we provide qualitative results in $\color{brown}{comparison\underline{}md\underline{}size.mp4}$ in the updated Supplementary Material.
>
> To determine the size of the motion dictionary, we tested 5, 10, 20, 40 and 100 motion directions. In case of a dictionary size of 20, we observe that generated results are of satisfying visual quality, achieving superior or equal quantitative results compared to SOTA. As shown in $\color{brown}{comparison\underline{}md\underline{}size.mp4}$, models with 5 and 10 directions are limited w.r.t. modeling complex motion. For instance, some motion patterns are clearly missing in the generated results. In the case of a motion dictionary endowed with 20 or more motion directions, animated results are of high quality, however the difference between models appears minor. Due to such observations, we select a motion dictionary comprising 20 motion directions.
>
> **Q2.2. How does M value affect the learned motion dictionary in terms of disentanglement and interpretability?**
>
> **A2.2.**  While more directions are beneficial in decomposing the original motion space of training data into significantly simpler local motion sub-spaces; it remains unclear whether simple local motion sub-spaces are able to provide more interpretability, as it is hard to objectively define interpretability. Currently, due to lack of tools for systematic analysis of semantics in the motion dictionary, we rely on the visualization of linear manipulation to interpret each direction. In future work, we plan to construct datasets, where different types of motions are represented and well-annotated, which would be instrumental for the study of interpretability.

---

> > ### Author Response · Authors · 2021-11-20
> > **Response from the Authors (part 2/2)**
> >
> > **Q3. In Figure 4, the proposed method doesn't seem to be able to replicate the driving video's pose very well. The paper should probably also compare with [Wang et al 2021a], which has an online demo.**
> >
> > **A3.** We note that our method is able to replicate motion correctly in Fig. 4, even if the head pose in the source image and driving video are substantially different. Fig. 4 might be misleading, as we illustrate the results using **relative motion transfer**, without displaying the first frame of the driving video (discussed in Sec. 3.4). In contrast, **absolute motion transfer** generally imposes the head pose of the driving video onto the source image. Descriptions have been added in the revised manuscript.
> >
> > Existing methods including ours, FOMM and MRAA all support both, absolute and relative motion transfer modes. In the updated Supplementary Material,  $\color{brown}{comparison\underline{}relative\underline{}absolute.mp4}$ is the video version of Fig. 4, where we show generated videos from different methods using both modes.
> >
> > We note that in **relative transfer mode**, as also mentioned in FOMM, source image and the first frame of a driving video are required to encompass a similar starting facial pose. In case that two images have large pose variations, the motion of a face in a driving video can be well transferred, however will start from the pose of source image. Since in our cross-dataset experiment, image-video pairs between FFHQ and VoxCeleb are randomly constructed, we do not have a pose-similarity constraint for each pair. Hence pairs, as shown in Fig. 4, might contain similar (last row) or substantially different poses (second row).
> >
> > As shown in $\color{brown}{comparison\underline{}relative\underline{}absolute.mp4}$, facial appearance is well-preserved in relative transfer mode, whereas absolute transfer might create artifacts, especially when source and driving images have large pose discrepancy. However, we observe that our method outperforms the SOTA also in **absolute transfer mode**.
> >
> > As [Wang et al 2021a] do not provide pretrained models, complete testing code and training data, we are not able to reproduce their results and conduct a fair comparison with their approach.

---

> > > ### Comment · Reviewer_EXdL · 2021-11-30
> > > **Thank you for the responses**
> > >
> > > Thanks for clarifying my concerns and providing additional materials. I think the additional material would be helpful for readers to understand the work to a fuller extent. In general, I appreciate the idea of the paper, and the results are encouraging. Although it remains a challenge to interpret the learned representations, I hold favorable views towards the paper and will maintain my initial rating after the discussion.

---

### Author Response · Authors · 2021-11-20
**To all reviewers**

We really appreciate all five reviewers for their careful reviews and valuable comments.
We have uploaded 10 additional videos in the updated Supplementary Material towards better answering reviewers' questions and demonstrating our results.

* $\color{brown}{vox256\underline{}linear\underline{}manipulation.mp4}$: motion dictionary visualization for VoxCeleb (256 resolution)
* $\color{brown}{vox512\underline{}linear\underline{}manipulation.mp4}$: motion dictionary visualization for VoxCeleb (512 resolution)
* $\color{brown}{ted256\underline{}linear\underline{}manipulation.mp4}$: motion dictionary visualization for TED-talks
* $\color{brown}{taichi256\underline{}linear\underline{}manipulation.mp4}$: motion dictionary visualization for TaichiHD
* $\color{brown}{comparison\underline{}relative\underline{}absolute.mp4}$: video version of Fig. 4. Comparison among LIA, FOMM and MRAA on VoxCeleb by using both relative and absolute motion transfer modes.
* $\color{brown}{realworld\underline{}vox512.mp4}$: real-world demos for 512 resolution talking head animation.
* $\color{brown}{comparison\underline{}md\underline{}size.mp4}$: comparison of LIA-models using five (5, 10, 20, 40, 100) motion dictionary sizes on three datasets.
* $\color{brown}{comparison\underline{}2encoders.mp4}$: comparison of models using (1) one single encoder and (2) two encoders without motion dictionary.
* $\color{brown}{comparison\underline{}orthogonal.mp4}$: comparison of models using orthogonal and non-orthogonal constraints in motion dictionary.
* $\color{brown}{comparison\underline{}mocoganhd.mp4}$: comparison between MoCoGAN-HD and our method.
* $\color{brown}{failure\underline{}cases.mp4}$: examples of failure cases from our method.

We have uploaded a revised manuscript incorporating reviewers' feedback. Below is a summary of the main changes:
* The reference image $x_r$ has been clarified in Sec. 3.1.
* Fig.4 has been updated to reduce misleading.
* We have shown quantitative results of using 40 and 100 motion directions in Tab. 5.
* We provide details of model architecture in App. A.
* We provide analysis of effectiveness of orthogonal basis in App. B.8.
* We discuss limitations of our method in App. B.9.
* We provide an ethic statement in App. C.

---

### Decision · Program_Chairs · 2022-01-20

**Decision:**

Accept (Poster)

**Comment:**

This paper proposes a self-supervised auto-encoder latent image animator that animates images via latent space navigation. The task of transferring motion from driving videos to source images is formulated as learning linear transformations in the latent space. Experiments conducted on real-world videos demonstrate that the proposed framework can successfully animate still images. The proposed framework is novel, the experimental results are supportive and promising. However, some related works are still missing and might need to be added to the current paper for discussion and comparison.

The rebuttal has addressed all major concerns raised by all 5 reviewers. The revised paper also included some feedback from the reviewers, except those discussions and comparisons with some missing related works pointed out by reviewers. After the rebuttal, all reviewers tend to accept the paper. AC agrees with the reviewers and recommends accepting the paper as a poster. Lastly, AC urges the authors to further improve their paper by incorporating the discussion on other missing related works suggested by the reviewers.